# Complexity of cortical wave patterns of the wake mouse cortex

Yuqi Liang [1,9], Junhao Liang [1,9], Chenchen Song [2,9], Mianxin Liu [1,3,9], Thomas Knöpfel [2,4,10] ✉, Pulin Gong [5,6,10] ✉ & Changsong Zhou [1,7,8,10] ✉

Rich spatiotemporal dynamics of cortical activity, including complex and diverse wave patterns, have been identified during unconscious and conscious brain states. Yet, how these activity patterns emerge across different levels of wakefulness remain unclear. Here we study the evolution of wave patterns utilizing data from high spatiotemporal resolution optical voltage imaging of mice transitioning from barbiturate-induced anesthesia to wakefulness (N = 5) and awake mice (N = 4). We find that, as the brain transitions into wakefulness, there is a reduction in hemisphere-scale voltage waves, and an increase in local wave events and complexity. A neural mass model recapitulates the essential cellular-level features and shows how the dynamical competition between global and local spatiotemporal patterns and long-range connections can explain the experimental observations. These mechanisms possibly endow the awake cortex with enhanced integrative processing capabilities.

In the living brain, electrical activity is always present, also in the absence of external stimuli. In early studies aimed at understanding how perception and behavior emerges from neuronal activities, spontaneous ("ongoing") activity was treated like "irrelevant noise" and removed by averaging over repeated trials[1,2]. However, the first pioneering electroencephalography studies already recognized the significance of spontaneous neuronal activities and its relation to brain states[3]. More recently, a large body of evidence supported the idea that a signature of recovery from general anesthesia to wakefulness or conscious states is an increasing complexity of cortical spontaneous activities[4,5]. These studies have mainly focused on the temporal dynamics in terms of correlated spontaneous activity across brain regions. For instance, it has been shown that physiologically reversible unconscious states such as sleep and anesthesia in humans and animals are characterized by decreased long-range correlations compared to conscious states[5–7]. However, the widely considered zero-lag correlations can only reveal the synchronization of brain activities while ignoring the complex neural dynamics unfolding both in space and in time that are associated with the propagation and processing of neuronal information.

Accumulating evidence showed that spontaneous cortical activity[8,9] as well as evoked activity[9–13] exhibit rich spatiotemporal patterns organized as traveling waves. Phase velocity fields (PVF) analysis, a method adapted from turbulence physics[14], is able to extract different types of wave patterns and quantify their features. Traveling waves can have different forms, including planar traveling waves[12,15], spiral waves that rotate around a central point[8], source and sink patterns that expand from or contract to a point[14,16], and saddle patterns[14]. These findings raise several fundamental questions: How are these spatiotemporal wave patterns affected by the general brain states (e.g. the anesthetized and awake states)? Which properties of these waves can be used as neural correlates of the recovery process from

[1]Department of Physics, Centre for Nonlinear Studies and Beijing-Hong Kong-Singapore Joint Centre for Nonlinear and Complex Systems (Hong Kong), Institute of Computational and Theoretical Studies, Hong Kong Baptist University, Kowloon Tong, Hong Kong. [2]Laboratory for Neuronal Circuit Dynamics, Imperial College London, London, UK. [3]Shanghai Artificial Intelligence Laboratory, Shanghai 200232, China. [4]Laboratory for Neuronal Circuit Dynamics, Hong Kong Baptist University, Kowloon Tong, Hong Kong. [5]School of Physics, University of Sydney, Sydney, New South Wales 2006, Australia. [6]ARC Centre of Excellence for Integrative Brain Function, University of Sydney, Sydney, New South Wales 2001, Australia. [7]Research Centre, Hong Kong Baptist University Institute of Research and Continuing Education, Shenzhen 51800, China. [8]Department of Physics, Zhejiang University, 38 Zheda Road, Hangzhou, China. [9]These authors contributed equally: Yuqi Liang, Junhao Liang, Chenchen Song, Mianxin Liu. [10]These authors jointly supervised this work: Thomas Knöpfel, Pulin Gong, Changsong Zhou. ✉e-mail: tknopfel@knopfel-lab.net; pulin.gong@syndey.edu.au; cszhou@hkbu.edu.hk

anesthesia to wakefulness? Which dynamical mechanisms underlie this process?

To address these questions, here we use mesoscopic high spatial coverage voltage imaging approach to monitor cortex-wide activity, which currently has only been achieved in mice. We then use the advanced PVF wave analysis method on mesoscopic voltage imaging data of the mouse cortex as the mice transitioned from barbiturate-induced anesthesia to wakefulness. We find that, when recovering from anesthesia to wakefulness, wave directions are less homogenous, and wave speeds are overall decreased while speeds of large waves are increased. We identify typical wave patterns including large-scale traveling waves, standing waves and complex local wave patterns like sources, sinks, and saddles at smaller spatial scales[17]. The principal modes of whole cortex-scale waves, defined as the singular value decomposition (SVD) modes of the PVFs, are similar in the anesthetized and wakeful states, but the awake cortex exhibited a smaller proportion of these principal modes and more local wave patterns with complex dynamics.

To explain these observations, we further develop a neural mass model that incorporates the key actions of barbiturates on the kinetics of chemical synapses[18,19] and the coupling strength of gap junctions[20]. We demonstrate that our model can reproduce the increased complexity of wave patterns that accompany approaching wakefulness as found in our experimental data. By performing dynamical stability analysis, we find that the strong competition between global and local activity states underlies the emergence of spatiotemporal complexity during the transition from anesthesia to wakefulness. In addition, we use the model to predict that long-range connections more efficiently transfer neural activities and information between cortical areas in the wakeful state. Together, our combined experimental and modeling studies reveal the essential features of spatiotemporal brain dynamics characteristics for wakefulness, and the dynamical mechanisms underlying these features.

## Results

### Characteristics of population activity waves depend on the level of wakefulness

We used mesoscopic optical imaging of mice expressing a genetically encoded voltage indicator in cortical pyramidal neurons, to access spontaneous population voltage activity across both hemispheres of the dorsal cortex. In one set of experiments ($N = 5$ mice), imaging began as the mice underwent light anesthesia induced by a bolus injection of pentobarbiturate, and continued until the mice woke up as indicated by occasional spontaneous coordinated whisker and body movements (SI Video 1 and 2). We termed the latter condition as "post woken". Another data set ($N = 4$ mice) was obtained from mice that were well habituated to the imaging conditions and that had been free of anesthesia for at least 3 days prior to the imaging session (SI Video 3). We termed this experimental condition as "fully awake".

In all three conditions, we observed a maximum frequency power of the cortex-wide population voltage activity at around 2 Hz corresponding to delta waves (1–4 Hz power on average contributes 70.2% (anesthetized), 67.2% (post woken), and 67.5% (fully awake)) (Fig. 1a, N: anesthetized = 5 mice, post woken = 5 mice, fully awake = 4 mice). Time-resolved power spectra indicated slow non-rhythmic waves of activity occurring at 1-4 Hz in both the anesthetized and post woken states (Fig. 1b). Frequency power is a classical measure of neuronal activity but it should also be noted that variation of periodic component of the power spectra on top of the aperiodic background with

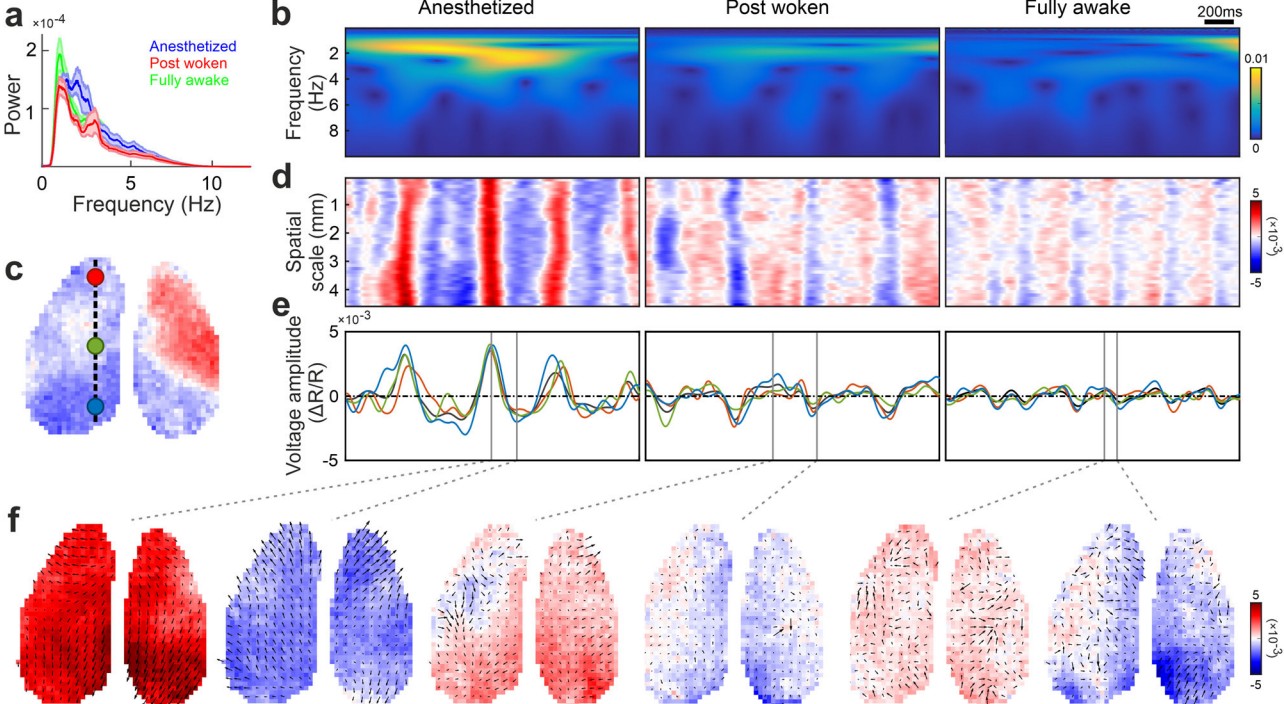

**Fig. 1 | Wakefulness alters cortical voltage activity patterns. a** Power spectrum of the average cortex-wide voltage activity at different brain states (mean+SEM. SEM is indicated by the shading. N: anesthetized = 5 trials, post woken = 5 trials, fully awake sample number = 19 trials). **b** Time-resolved power spectrum of voltage signals at different brain states. Voltage data taken from the position marked by the blue spot in **c**. Warmer color represents higher power. **c** Location of three arbitrary spots on the cortex mask. The background represents the voltage amplitude at one random time point; colorbar same as **d**. **d** Example voltage activity along an arbitrarily chosen frontal-posterior line on the left hemisphere as indicated in **c** (circles on dashed line). **e** Voltage activity from three representative locations indicated in **c**. The black trace is the cortex-wide spatially averaged voltage signal. **f** Examples of phase velocity fields at selected time points. The arrow orientation indicate wave propagation direction and arrow length indicates propagation speed. All data presented here are comparison between anesthetized, post woken and fully awake states from a representative mouse.

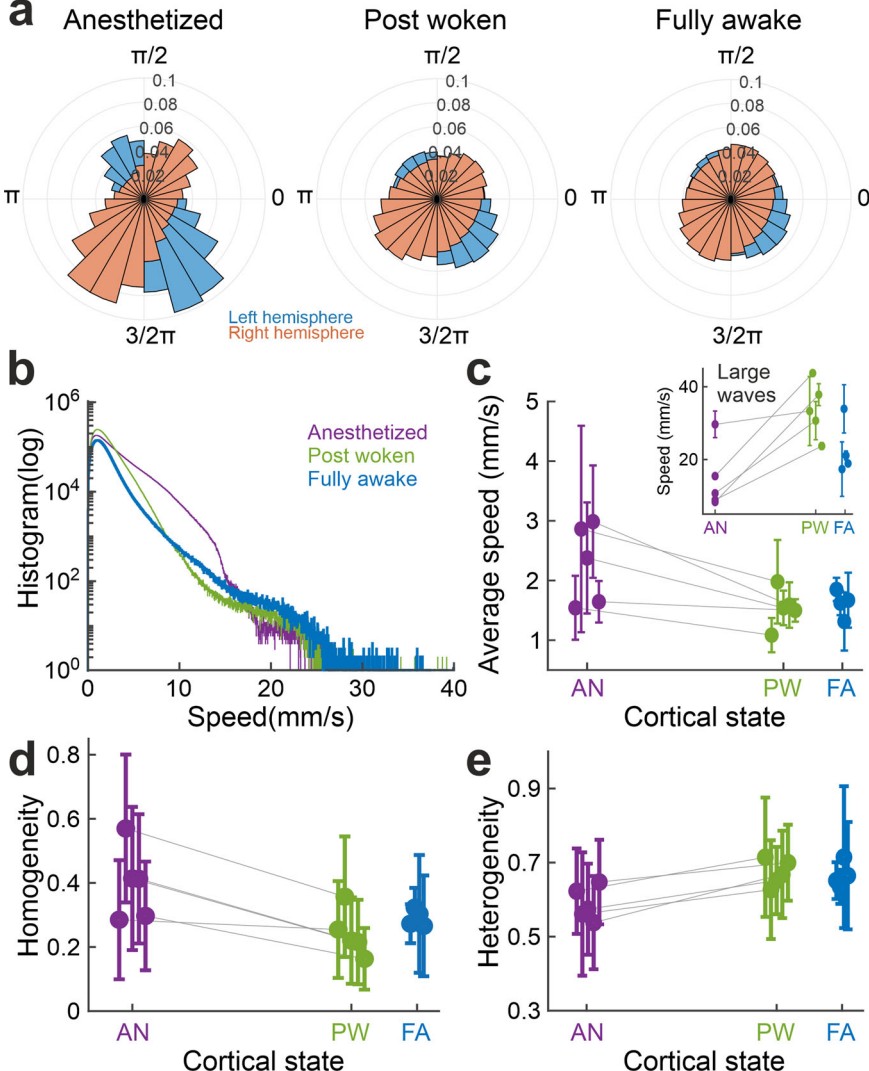

**Fig. 2 | Wakefulness alters the direction and speed of the waves. a** Angle histogram of wave propagation directions for each hemisphere over one 3-min trial from a representative mouse at different brain states. **b** Distribution of the instantaneous wave propagation speed over the trials in **a**. **c** Average speed of PVFs at different brain states. Inset: Average speed of large waves. **d** Homogeneity of wave direction at different brain states. **e** Heterogeneity of wave speed at different brain states. **a-b**: from one representative mouse. **c-e**: each datapoint represents one mouse (anesthetized (AN) and post woken state (PW): 5 mice; fully awake state (FA): 4 mice; gray line connects data from the same mouse). For **c, d** and **e**, data are presented as mean values +/− SEM, SEM is indicated by error bars.

brain states has been previously observed[21]. We observed spatiotemporal patterns in the voltage maps (Fig. 1c, d), with activity waves that exhibited larger oscillation amplitudes during the anesthetized state (Fig. 1e). We then characterized these patterns using PVF analysis. We detected continuous and complex wave propagation in all brain states (see Methods), but the wave patterns in the anesthetized state were more coherent (Fig. 1f). We then further quantitatively characterized these observed differences in the wave patterns.

We first quantified the propagation direction and speed of the detected waves. The voltage waves propagated in preferred directions in all three brain state conditions, but the wave directions showed a broader distribution in the post woken and fully awake states than in the anesthetized condition (Watson-Wheeler test, $W(4) = 7.111e5$, $p < 0.001$, for both comparisons. N = 32385963 for each condition) (Fig. 2a). The wave propagation speed distributions are also state-dependent (Friedman's test, chi-square(1)=1.268e6, $p < 0.001$, Kendall's $W = 0.495$, 95% confidence intervals = [0.495,0.495] for anesthetized and post woken and chi-square statistic(1)=3.497e6, p < 0.001, Kendall's $W = 0.504$, 95% confidence intervals = [0.504,0.505] for anesthetized and fully awake. N = 35616483 for each condition) with a

larger contribution of faster waves in the post woken and fully awake states than in the anesthetized condition (Fig. 2b). From anesthetized to post woken, the average speed of PVF decreased (Fig. 2c, one-sided Wilcoxon signed rank test, $W = 15$, $p = 0.0312$, Hedges' $g = 1.230$, 95% confidence intervals = [0.818, 3.928]) even though the average propagation speed (20-40 mm/s) of large waves increased (Fig. 2c inset, one-sided Wilcoxon signed rank test, $W = 0$, $p = 0.0312$, Hedges' $g = -2.026$, 95% confidence intervals = [−6.014, −1.151]) for all mice analyzed (N = 5 mice for both conditions). The trends remain largely consistent when comparing anesthetized mice with the set of fully awake mice but without reaching statistical significance (on average speed: one-sided Wilcoxon rank sum test, $U = 29$, $p = 0.2063$, Hedges' $g = 0.965$, 95% confidence intervals = [0.029, 3.986]; on large wave speed: one-sided Wilcoxon rank sum test, $U = 18$, p = 0.0556, Hedges' $g = -0.877$, 95% confidence intervals = [−5.660, 0.189]. N = 4 mice for fully awake). The presence of large waves with faster speed is consistent with the bumps around 20−30 mm/s in the speed distributions in Fig. 2b derived from PVF. The average propagation speed of these large waves is consistent with those found in earlier studies[22,23]. However, we did find that those localized complex wave patterns have much slower phase velocities

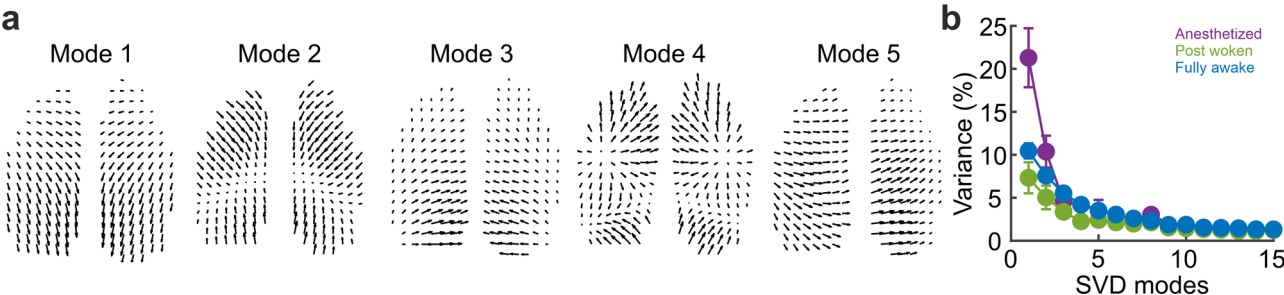

**Fig. 3 | Principal modes of the phase velocity fields and their contributions under three states. a** Top 5 modes obtained by the singular value decomposition of all phase velocity fields (SVD variance of mode 1: 16.5%, mode 2: 9.8%, mode 3: 5.4%, mode 4: 4.2%, mode 5: 3.9%; Top 5 modes total: 39.8%, each vector for each second pixel shown). **b** Variance distribution of the top 15 modes at different brain states (mean +/−SEM, SEM is indicated by error bars, anesthetized $N = 5$ trials, postwoken $N = 5$ trials, fully awake = 19 trials).

(distribution peaked around 3 mm/s; Fig. 2b). The apparent slow average PVF speed of these local waves is, in part, due to the fact that phase singularities at the center of the complex waves are characterized as zero speed by nature. Another factor likely limiting the phase propagation of localized complex waves are the constrains imposed by their surrounding activity patterns.

Next, we used homogeneity to measure the coherence in the wave direction. We consistently found a decrease (statistically significant for anesthetized vs post woken, one-sided Wilcoxon signed rank test, $W = 15$, $p = 0.0312$, Hedges' $g = 1.378$, 95% confidence intervals = [0.900, 3.747]; not reaching statistical significance for anesthetized vs fully awake, one-sided Wilcoxon rank sum test, $U = 31$, $p = 0.0952$, Hedges' $g = 0.961$, 95% confidence intervals = [0.200, 2.660]. $N = 5$ mice for anesthetized and post woken, $N = 4$ mice for fully awake) in homogeneity from anesthetized to post woken and fully awake states, indicating that wave propagation directions are more disordered in the post woken and fully awake states (Fig. 2d). Heterogeneity measurements, which reflect speed variance, slightly increased with the transition to the awake states (Fig. 2e, anesthetized vs postwoken, one-sided Wilcoxon signed rank test, $W = 0$, $p = 0.0312$, Hedges' $g = -2.065$, 95% confidence intervals = [−5.384, −1.680]; anesthetized vs fully awake, one-sided Wilcoxon rank sum test, $U = 15$, $p = 0.0079$, Hedges' $g = -2.178$, 95% confidence intervals = [−6.644, −1.542]. $N = 5$ mice for anesthetized and post woken, $N = 4$ mice for fully awake).

## Principal components of phase velocity fields

We evaluated how the spatiotemporal features of the activity waves changed with brain state by determining the SVD of the phase velocity fields. The standard principal SVD modes obtained from all mice and states represent the typical wave propagation trajectories (Fig. 3a, $N = 5$ mice for anesthetized and post woken (5 trials per condition), $N = 4$ mice for fully awake (19 trials)). Mode 1 represents a large-scale propagating wave across the dorsal cortex (both hemispheres), and its contribution to the overall PVF variance decreased sharply from the anesthetized to the post woken state (Fig. 3b). This is consistent with the notion that large-scale brain dynamics is low dimensional in the anesthetized state[24] and is dominated by the first mode. The top 5 modes together occupied a large portion of PVF variance during the anaesthetized state (average 48.1%) but contributed much less to the awake state (post woken average 22.3%, fully awake average 30.2%) (Fig. 3b, anaesthetized vs post woken one-sided Wilcoxon signed rank test, $W = 15$, $p = 0.0313$, Hedges' $g = -2.026$, 95% confidence intervals = [1.6361, 4.8926], anaesthetized vs fully awake, one-sided Wilcoxon rank sum test, $U = 100$, $p = 0.0043$, Hedges' $g = 1.589$, 95% confidence intervals = [0.611, 3.381], $N = 5$ trials each for anesthetized and post woken, $N = 19$ trials for fully awake). This increased contribution of other smaller modes in the awake states indicate increased wave complexity, suggesting that the wave patterns become more localized and diverse,

consistent with our earlier observations from amplitude patterns and PVF patterns (Fig. 1d-f).

## A change of wave patterns with the transition to wakefulness

Next, we classified the detected wave patterns into plane wave, standing wave, source, sink and saddle waves (see Methods; Fig. 4a). A plane wave indicates coherent large-scale propagation, while a standing wave indicates an almost synchronous non-propagating large-scale activity. Source, sink and saddle waves typically occupy small areas, hence are referred to as local wave patterns (Supplementary Fig. 1). The effects of filtering on the complex wave detection are explored in Supplementary Fig. 2. We also calculated the curl and the divergence based on the PVF to complementarily illustrate the wave patterns (Supplementary Fig. 3).

Source, sink and saddle patterns can co-exist on the same PVF frame[17]. If none of these wave patterns were detected in a given frame, we defined such a frame as unclassified. With the transition into the wakeful state, we observed that the number of sources, sinks and saddles largely increased, while plane waves and standing waves decreased (Fig. 4b, $N = 5$ mice for anesthetized and post woken (5 trials per condition), $N = 4$ mice for fully awake (19 trials)).

The increase in the occurrence probability of local wave patterns can be caused by an increase in either the duration or the number of wave patterns, or both. During anesthesia, large-scale coherent patterns like plane waves and standing waves typically sustained for longer time than the complex local wave patterns, but the frequency of their occurrence remained low across states (Fig. 5a, b). The number of total wave patterns detected in post woken and fully awake conditions increase by a factor of 1.4–2.3 as compared to the anesthetized states (Fig. 5c), while the occurrence of plane waves and standing waves decreases with transition into wakefulness (Fig. 5b). In the fully awake sessions, we observed phases with lowered heartbeat frequency, reduced facial movements and reduced occurrence of complex local waves (Supplementary Fig. 4) and interpreted these periods as transient transitions into a sleepiness state from a state of alertness.

## Neural mass model of wave dynamics

Our experimental data demonstrated that the transition to wakefulness is associated with more local wave patterns and with increased complexity (Figs. 1–5). We then used a neural mass model to elucidate the possible neural mechanisms of these changes in the wave dynamics.

The model represents a cortical sheet with coupling between cortical tissue voxels (see Methods). Each cortical voxel contains local excitatory and inhibitory neuron populations that interact through chemical synapses with AMPA and GABAa receptors and electrical synapses (gap junctions). We modeled the effect of barbiturate anesthesia on AMPA and GABAa receptors and gap junctions[19,25,26], with an overall effect of increased inhibition that can lead to a reduction of

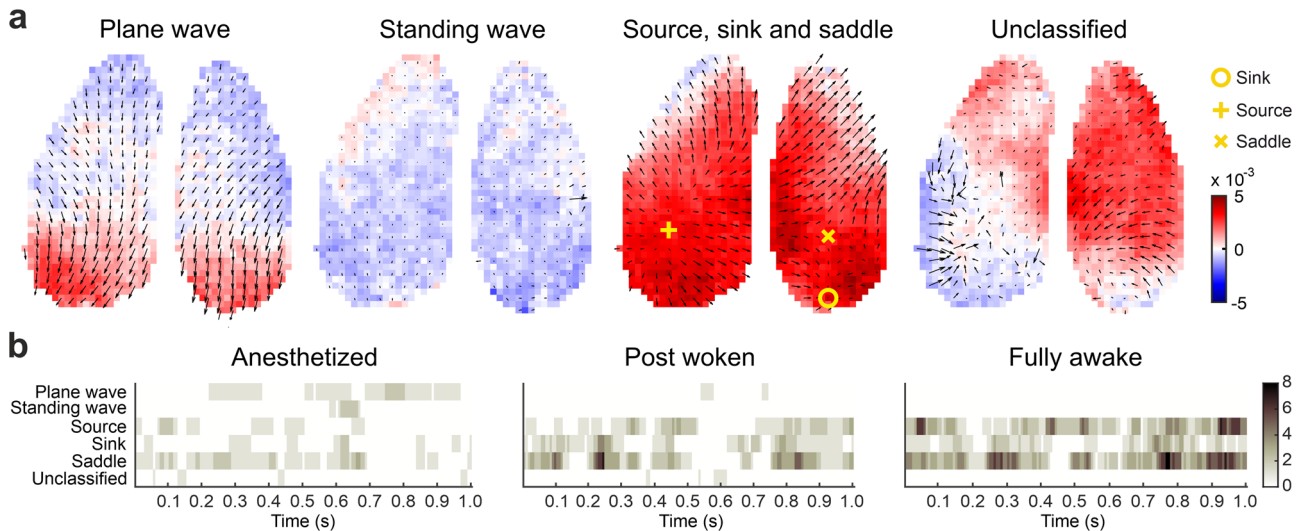

**Fig. 4 | Classification and detection of specific wave patterns at the cortex-wide scale. a** Examples of wave patterns: plane wave, standing wave, source, sink and saddle. The background color represents the voltage amplitude. **b** Time course of the wave patterns from example trials under anesthetized (left), post woken (middle) and fully awake (right) state. Color bar shows the number of waves at the time of a snapshot.

local circuit excitability (i.e. more difficult to be excited by excitatory synaptic drive). We modeled the dynamics of population neuronal membrane potential by neural field equations with diffusion operator, which effectively captures the strong near-neighbor connections of canonical cortical circuits. The coupling between cortical voxels results from currents through gap junctions and chemical synaptic receptors.

We first considered only the local coupling without long-range synaptic connections, and tested whether the model can account for the features observed in the wave pattern transitions in the experimental data. We replicated the same analysis as in the experimental data while changing the parameter controlling the degree of anesthetic effect (i.e., parameter $p$ in Eq. (10) in Methods). With deeper anesthesia, activity waves typically have larger sizes and longer propagating distances (Fig. 6a), which are consistent with smaller wave number/larger wavelength (Supplementary Fig. 7). The average speed of PVFs decreased during the transition from anesthetized to wakeful states (Fig. 6b). Our simulation also showed an overall increased heterogeneity (Fig. 6c) and decreased homogeneity (Fig. 6d) of the PVFs through transitioning from the anesthetized to wakeful states. Further, the number of complex local waves increased with a reduction of the barbiturate effect (Fig. 6f, g), similar to the trend we observed in the experimental data as mice transitioned from anesthetized towards awake state. However, the average duration of these complex waves was relatively stable (Fig. 6e), which is also similar to the observation in our experimental data (Fig. 5a). Thus, our numerical simulation results based on a minimalistic model reproduce the key dynamic features during the transition from anesthetized to awake state.

To obtain a deeper theoretical understanding of these brain state-dependent dynamical mechanisms, we performed a linear perturbation-based stability analysis (see Supplementary, Figs. 5, 6), involving Hopf instability (i.e., the tendency of global neural state oscillations) and Turing instability (i.e., the tendency of the formation of static locally distributed spatial patterns). We made two observations. First, in the anesthetized state, Hopf instability dominates Turing instability, hence more large-scale propagating waves would naturally emerge in this state. Second, in the wakeful state, there is a strong competition between Hopf and Turing instabilities, which induces a superposition of local spatial patterns and global state oscillations, causing smaller and more localized wave patterns. Theoretical analysis also provided an explanation for the reduction of the average wave speed (Mann-Kendall

test, $Z = -2.6301$, $p = 0.0085$), while large waves increased in propagation speed (Mann-Kendall test, $Z = 2.6301$, $p = 0.0085$; Fig. 6b inset).

When considering only the interactions between adjacent pixels, an increase of neural excitability is sufficient to change wave patterns similar to observations in vivo during the transition into wakeful state. However, the physiological awake state is also characterized by specific patterns of inter-area correlations (including anti-correlations), such as the canonical resting-state functional connectivity[27,28]. These patterns are shaped by inter-area structural connections between cortical areas. We therefore explored the effect of adding inter-area connections on the emergence of wave patterns in our neural mass model. In an ideal case of a noise-free system with only one weak inter-area connection, we observed that the wave activities can transmit through the inter-area connection only in the awake state, whereas the lower excitability in the anesthetized state does not support the transmission through the inter-area connection (Supplementary Fig. 8).

To examine this feature in a more biologically plausible scenario, we then constructed unidirectional region-to-region connections, with the projections randomly distributed between pairs of voxels from two regions of equal sizes (both $30 \times 30$). We simulated spontaneous cortical activity with the presence of noise, and used signal coherence to quantify the information transfer between two cortical regions[29,30] (see Methods). Our model suggested that lower frequency bands have higher coherence, which is caused by large coherent waves, while higher frequency bands have lower coherence which reflects that the disordered local waves dominated the overall dynamics. When the two regions have weak inter-area connections, the activities of the two regions show greater coherence in the anesthetized state in the frequency band examined (0.5–12 Hz as in the data; Fig. 7a). However, if the inter-area connections are strengthened, the coherence at the wakeful state increased (Fig. 7a).

To test this prediction derived from our model, we applied the corresponding analysis to our experimental data. We first selected two pairs of cortical regions with similar area sizes and distances but different connection strengths (weak inter-area connection: from SSp-bfd to RSPagl; strong inter-area connection: from SSp-bfd to Mop; selected according to connection strengths provided by the Allen Mouse Brain Atlas[31]), and used the experimental data from these pairs of regions at different brain states to compute the activity coherence. Consistent with the model results, the activity coherence between regions with strong inter-area connections is larger than the coherence between

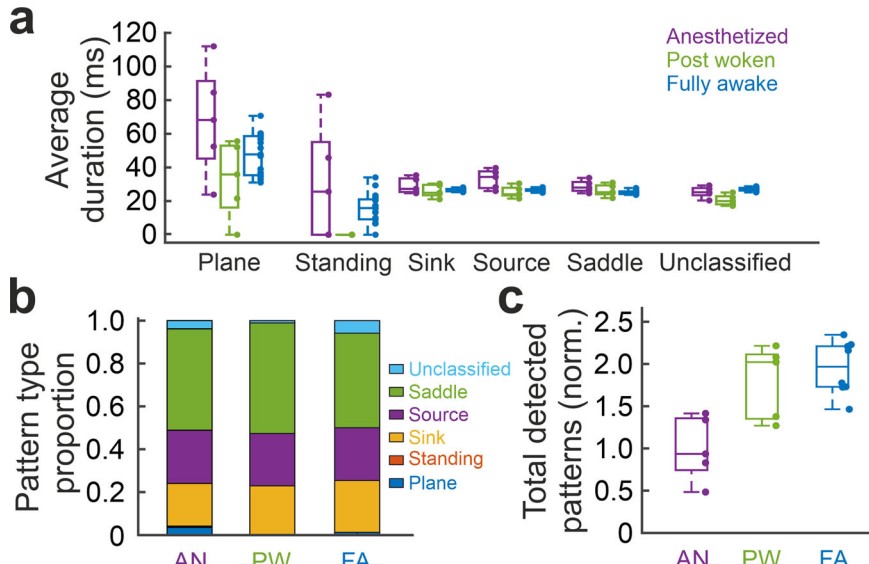

**Fig. 5 | Spatiotemporal dynamics of wave patterns. a** Boxplots for average duration of the different wave patterns at different brain state conditions. **b** Proportion of the different wave patterns at different brain state conditions. **c** Normalized total detected pattern numbers at different brain states. AN: anesthetized, PW: post woken and FA: fully awake. For AN, total pattern numbers for each trial are normalized to the mean total pattern number across trials at AN state.

For PW and FA, total pattern numbers for each trial are normalized to the total pattern number of the corresponding AN state. **a** and **c**: for boxplots, center line indicates the median, lower and upper limits of the box indicate the 25% and 75% percentiles, and whiskers give the maximum and minimum values. Outliers are indicated with "+". Dots aside the boxplots are showing data from anesthetized $N = 5$ trials, post woken $N = 5$ trials, fully awake $N = 19$ trials for **a** and $N = 10$ for **c**.

regions with weak inter-area connection (Fig. 7b). Comparing with the anesthetized state, coherence under the awake state is discernibly increased (Fig. 7b). For statistical analysis we selected six additional pairs of regions (Supplementary Table 1) provided by the Allen Mouse Brain Atlas[31], and computed the signal coherence on the experimental data from these regions (N = 5 mice).

From our above observations, we also noted that coherence is differentially affected by the presence of inter-area connections at different frequency ranges. Therefore, we further examined activity coherence of the model at different frequency bands (0.5–4 Hz, 4–8 Hz, 8–12 Hz) corresponding to delta, theta, and alpha brainwave bands respectively (Fig. 7c). In the absence of inter-area connections, there is little coherence at either anesthetized or awake states, indicating that wave propagation is mediated by successive neighboring connections and, hence, cannot transfer information effectively and rapidly between two distant regions (Fig. 7c). As the strengths of the inter-area connections increase, the coherence under the awake state increased (two-sided Wilcoxon signed rank test, 0.5-4 Hz: W = 1, $p = 6.14e\text{-}51$, Hedges' $g = -2.132$, 95% Confidence Interval = $[-2.298, -1.998]$, 4-8 Hz: W = 76, $p = 1.31e\text{-}50$, Hedges' $g = -1.844$, 95% Confidence Interval = $[-1.999, -1.713]$, 8–12 Hz: W = 13, $p = 6.93e\text{-}51$, Hedges' $g = -2.132$, 95% Confidence Interval = $[-2.301, -1.986]$, $N = 300$ trials) whereas the coherence under the anesthetized state does not show an increase (Fig. 7c). This confirms that the inter-area connections play less significant role in the anesthetized state due to lower local excitability in the circuits.

Similar analysis on the experimental data confirmed that the average coherence increased with the strength of inter-area connections at both states, but more prominently in the awake than the anesthetized state (Fig. 7d). This is consistent with the idea that inter-area projections play important roles in corticocortical information transmission in the wakeful than in the anesthetized state.

## Discussion

Identifying neural signatures underlying the absence or presence of conscious wakefulness is a long-standing problems in science[32]. In this study, by combining empirical observation and modeling, we have found a set of signatures for the transition from anesthetized to wakeful states in mice based on travelling waves in the cortex - including the increased spatiotemporal complexities of localized wave patterns - and outlined the possible underlying mechanism of these neural signatures. These signatures have far more complex spatiotemporal dynamics than expected by conventional views of temporal correlations[4,5] or attractors[33], and may have implications for understanding conscious processes of brain functions.

Our analysis of spatiotemporal wave-based signatures reconcile and extend key previous findings on the transition process from anesthetized to awake states[4,5,33]. We found that, during the anesthetized state, cortical dynamics are dominated by cortex-wide traveling waves. These global waves mainly propagate along the fronto-lateral to parietal-medial direction, largely constrained by the underlying structural network connectivity and anatomical gradients[17]. Cortex-wide waves give rise to positive correlations between activities among cortical areas. Our finding is thus consistent with the recent observations that following a loss of conscious wakefulness, coordinated brain activity is largely restricted to positive interareal correlations[4], but our findings also exceed these observations by showing that global waves cause such correlations.

With the transition to wakefulness, the proportion of large-scale waves decreases, and wave patterns become more localized and diverse. These increased spatiotemporal complexities of brain dynamics are generally consistent with the idea that there is a higher prevalence of a complex configuration of interareal interactions during conscious states, as quantified by temporal correlations[4,5]. However, by also taking space into account, our study has revealed a more comprehensive and reliable set of signatures of wakefulness, such as the increased speed of large waves and heterogeneity, and coexisting localized wave patterns. Wave patterns can naturally communicate information due to their propagation property[13,34]. The increased large wave speeds during the wakeful state can thus speed up wave-based communications between different brain areas, facilitating an efficient cortex-wide coordination. Furthermore, it has been proposed that interactions of localized neural waves can carry out a type of collision-based dynamical computation[34]. Our findings of multiple, coexisting

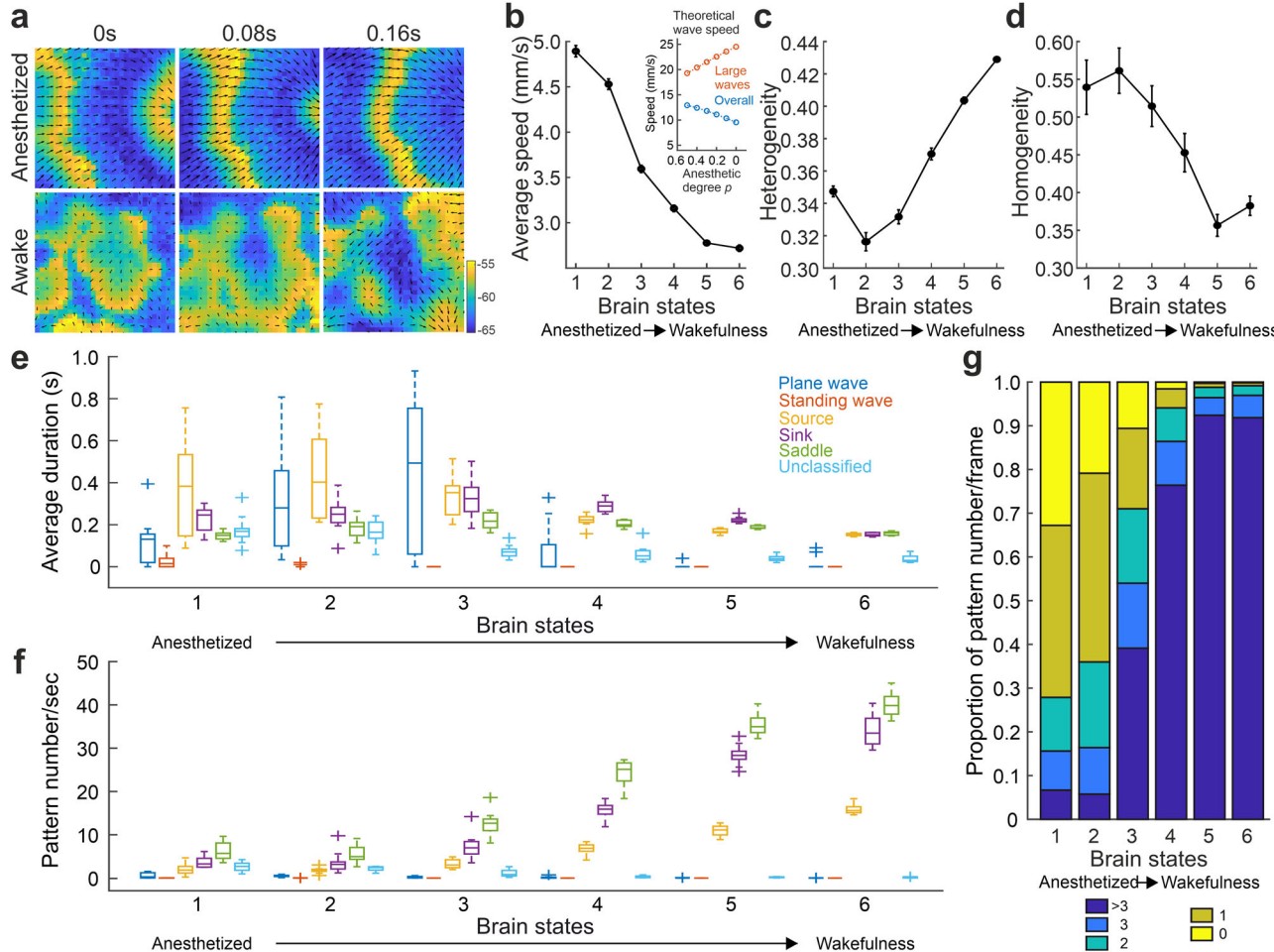

**Fig. 6 | State-induced changes of wave patterns in neural mass model.**
**a** Examples of simulated wave patterns. Color indicates the value of voxel voltage. Upper: anesthetized state (model parameter $p = 0.5$). Lower: awake state (model parameter $p = 0$). **b-g**, Spatiotemporal properties during waking up. Brain states 1-6 correspond to parameter $p = 0.5, 0.4, 0.3, 0.2, 0.1, 0$ in our model simulation, from anesthetized (state 1) to wakefulness (state 6). Error bar = SEM; $N = 10$ trials. **b** The average speed of waves. Inset: analytical prediction (see SI) of the overall and large-scale wave speeds from anesthetized to wakefulness states. Note the increase in the speed of large waves (inset) but decrease in the overall wave speed with the

transition into wakefulness (see Supplementary Information). c Heterogeneity of the wave speeds. **d** Homogeneity of the wave directions. **e** Boxplots for average duration of the different wave patterns. **f** Boxplots for average number of wave patterns every second. For boxplots, center line indicates the median value, lower and upper limits of the box indicate the 25% and 75% percentiles, and whiskers give the maximum and minimum values. Outliers are indicated with "+". $N = 10$ trials. **g** The probability of the number of local complex waves (source, sink and saddle) in each frame.

localized wave patterns and their interactions during the wakeful state thus suggest that they might enable such dynamical computation to be implemented in a fundamentally distributed and parallel manner. In the future, it would be interesting to study whether and how this wave-based distributed dynamical computation underlies cognitive functions with specific conscious experience.

The presence of localized wave and global wave modes found in our study is also consistent with theories, which propose that the conscious state possesses both global integration and a rich, complex repertoire of local functional states[35]. In our framework of wakefulness states, the global waves are ideal for implementing this global integration due to their relatively large sizes and large-scale propagations across the whole cortex. Further, these waves might enable the global "ignition" of a widespread network of regions; such an ignition has been proposed to be essential for conscious processing[36]. On the other hand, in our framework, each of the localized wave patterns is the neural substrate for exploring or sampling a rich repertoire of functional brain configuration states through its variable and hetero-geneous dynamics[37]. Finally, it is worth noting that as demonstrated in our study, global and localized waves may strike a right balance in terms of their occurrence rates for efficiently balancing both

integration and segregation, which is another requirement of con-scious states[35].

Limited by the practical resolutions in either space or time, human studies using EEG and fMRI often characterize neural spatiotemporal activities with temporal variability (e.g., entropy measure of com-plexity) or correlation (e.g., functional connectivity). It has been observed that increased entropy and decreased functional con-nectivity are associated with the emergence of consciousness from sleeping to wakeful state in humans[38–40]. A comparison between heal-thy wakeful human subjects and patients with disorders of con-sciousness further suggested that conscious cognition could be associated with long-distance coordination and high modularity in functional connectivity[5]. The current study suggests that these observations in humans could potentially be understood under the travelling wave framework, as the relative dominance of local events could cause increased entropy while reduced global waves could result in decreased functional connectivity. However, caution should be exercised when generalizing the results from mice to humans[41].

To understand the circuit-level mechanism behind the state transition, we used a neural mass model. The model highlights how increased excitability (of both excitatory and inhibitory neurons) in the

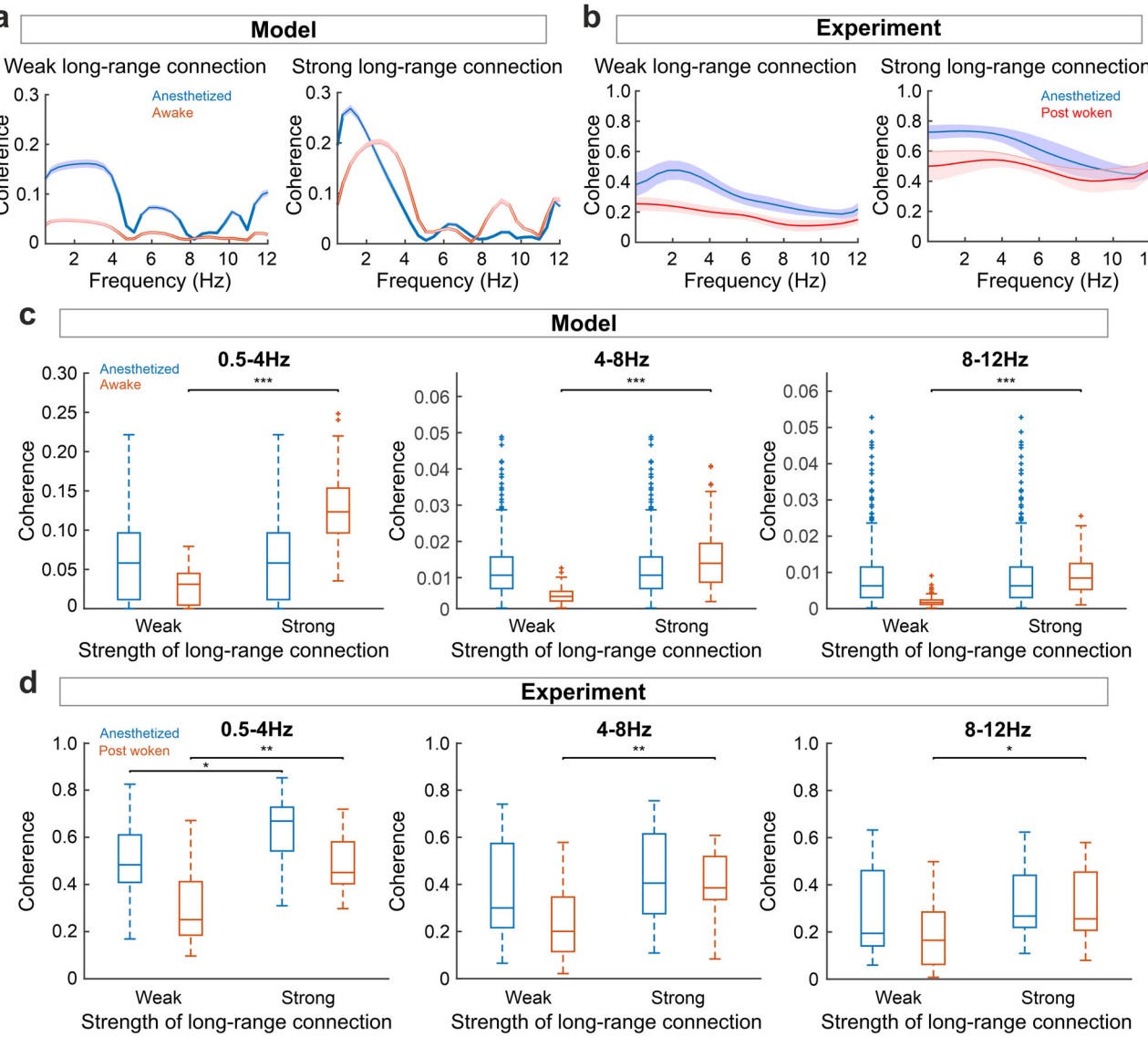

**Fig. 7 | Long-range connections enhanced the coherence both in anesthetized and awake state. a** Activity coherence between two cortical regions with weak and strong long-range connections in the anesthetized and awake states in the model simulation (mean + SEM. SEM is indicated by the shading. Both *N* = 100 trials). **b** Same as **a**, for experimental data (mean + SEM. SEM is indicated by the shading. Anesthetized state *N* = 5 trials, post woken state *N* = 5 trials). **c** Boxplots for activity coherence at different frequency bands between weakly (group of connection strength = 0, 5, 10) and strongly (group of connection strength = 20, 25, 30) connected cortical regions in the anesthetized and awake states in the model simulation. For boxplots, the center line indicates the median, the lower and upper limits of the box indicate the 25% and 75% percentiles, and the whiskers give the maximum and minimum values. Outliers are indicated with "+". Both *N* = 300 trials.

Two-sided Wilcoxon signed rank test was applied, resulting in 0.5–4 Hz: W = 1, *p* = 6.14e-51, Hedges' g = −2.132, 95% Confidence Interval = [−2.298,−1.998], 4–8 Hz: W = 76, *p* = 1.31e-50, Hedges' g = −1.844, 95% Confidence Interval = [−1.999,−1.713], 8-12 Hz: W = 13, *p* = 6.93e-51, Hedges' g = −2.132, 95% Confidence Interval = [−2.301,−1.986], *p < 0.05, **p < 0.01, ***p < 0.001. **d** Same as **c**, for experimental data (anesthetized state N = 5 trials, post woken state *N* = 5 trials). Two-sided Wilcoxon signed rank test was applied, resulting in 0.5-4 Hz: W = 104, *p* = 0.0103, Hedges' g = −0.707, 95% Confidence Interval = [−1.410,−0.282] and W = 111, *p* = 0.0020, Hedges' g = −1.069, 95% Confidence Interval = [−2.090,−0.526], 4-8 Hz: W = 106, *p* = 0.0067, Hedges' g = −0.902, 95% Confidence Interval = [−1.793,−0.333], 8-12 Hz: W = 103, *p* = 0.0125, Hedges' g = −0.776, 95% Confidence Interval = [−1.548,−0.184]. *p < 0.05, **p < 0.01.

wakeful state can empower long-range connections to initiate additional waves. Despite this increase in wave initializations, network stability is preserved due to their destructive interference. The model includes synaptic and cellular effects of barbiturate anesthesia, but we shall note that in the real cortex and under physiological conditions, increased excitability during the conscious state are likely to be facilitated by endogenous neurostimulators such as acetylcholine and amines[42,43].

The higher excitability state empowers both local processing and efficient signal transmission through long-range connectome. Our linear stability analysis showed that the transition from a dominance of Hopf instability in the anesthetized state to a strong

interaction between Hopf and Turing instabilities in the awake state caused the increased speed of large-scale traveling waves and the decreased speed of overall wave patterns. Besides the dynamical stability property due to the interaction of local excitability and neighboring coupling, the complex long-range connections in the cortical connectome make additional contributions to the emergence of complex wave patterns, especially prominent in the wakeful cortical state. Both effects further enhance and balance local and distributive processing. Thus, higher excitable states during wakefulness allow a more efficient propagation, processing and integration of information, in order to facilitate higher cognitive process and executive outputs.

## Methods

### Ethical statement

All experimental procedures were performed at Imperial College London UK in accordance with the United Kingdom Animal Scientific Procedures Act (1986), under Home Office Personal and Project Licenses following appropriate ethical review by Imperial College London Animal Welfare Ethical Review Body and Home Office.

### GEVI-based optical voltage imaging

We used CaMK2A-tTA;tetO-chiVSFP transgenic mice[44] that express the genetically encoded voltage indicator chimeric VSFP Butterfly[45] in pyramidal neurons across all cortical layers. We used the epi-fluorescence imaging approach that restricts optical access and signal detection to the superficial cortical layers (layers 1-3). Mesoscopic transcranial voltage imaging datasets were acquired following previously described methods[44,46–48]. Briefly, adult mice (aged 2-3 months of either sex) underwent isoflurane surgical anesthesia, and were implanted with a transcranial cortex-wide window on a thinned but otherwise fully intact skull and a head-fixation plate. After at least one-week post-operative recovery, animals were habituated to the dual-emission wide field epifluorescence macroscope used for image acquisition. Images were acquired with two synchronized CMOS cameras, using a high power halogen lamp for fluorescence excitation (Moritex, BrainVision) and the following optics (Semrock): mCitrine (donor) excitation 500/24, mCitrine emission FF01-542/27, mKate2 emission BLP01-594R-25, excitation beam splitter 515LP, and detection beam splitter 580LP. Spontaneous population membrane voltage fluctuations of pyramidal neurons were monitored as the ratio of changes of gain-equalized[47] fluorescence intensities at the two fluorescence emission wavelength bands.

Datasets were acquired over cortex-wide two-hemisphere fields of view at 150 Hz acquisition frame rate and 29 um × 29 um spatial resolution. During the imaging sessions (several trials of 180 s duration each), mice were initially anesthetized (induced by a bolus injection of 30 mg/kg pentobarbital sodium), and then allowed to recover over a state of sedation to wakefulness. Heart rate was used to monitor the state of the animal during this progression. We considered a brain state with the absolute lack of spontaneous limb and whisker movements as "anesthetized". We analyzed trials over five mice (Mouse 1-4 are males and Mouse 5 is female), and each mouse provided several trials from anesthetized condition to awake condition (mouse 1: eighteen trials; mouse 2: seven trials; mouse 3: three trials; mouse 4: three trials; mouse 5: six trials). We used one trial as "anesthetized" and one trial as "post woken" from each recovery mouse imaging experiments. Thus, we have a total of 5 trials for anesthetized and 5 trials for post woken for analysis. In addition, four mice were analyzed in the fully awake state (mouse 1: ten trials, mouse 6: seven trials, mouse 7: one trial, mouse 8: one trial). Each trial recorded spontaneous voltage activity for 180 s continuously.

### Behavioral monitoring

To monitor facial expression and forelimb movements, an additional camera equipped with a f25 mm lens was frame-synchronized with the brain imaging cameras and was directed towards the mouse's face. The scene was illuminated by infrared light (850 nm) to avoid interference with voltage imaging using the blue-red wavelengths range (<700 nm).

### Data preprocessing

We extracted the voltage signal from raw fluorescence signals[23,44,47], and applied 2-times coarse graining using bi-cubic interpolation (weighted average of pixels in the nearest 4-by-4 neighborhood; function *resize*, MATLAB, Mathworks Inc, USA) to obtain a 44 × 52 matrix to reduce the spatial noise before computing the phase velocity fields. Then, we applied 0.5-12 Hz (including delta, theta and alpha brainwave bands) bandpass filtering on the data (Chebyshev Type II,

function *filtfilt*, MATLAB, Mathworks Inc, USA). Periods with large temporal fluctuations (amplitude > 3 standard deviation) of the filtered voltage signals averaged over the field of view were defined as movement artifacts and were excluded from further analysis. Registration of the cortex-wide voltage imaging data into the Allen Mouse Brain atlas was performed as previously described[31].

To measure the behavioral states of the animals, we adapted the motion energy (ME) method[49] to analyze the facial expression movies synchronized to the brain imaging data. ME quantifies the amount of movements at a specific location of the image as the absolute intensity of the differences between the consecutive time points (frames) of the same pixel. We used the spatial average of the ME sequences to quantify overall motion.

### Phase velocity field

The phase velocity field (PVF) analysis used to characterize the cortex-wide spatiotemporal voltage patterns is adapted from physical theories of turbulence[9,14]. Briefly, this method is based on the assumptions that the contours (isolines) of the phase of brain oscillations move monotonically spatiotemporally. We used generalized phase[50] to avoid phase distortions on narrow band filter and extract appropriate phase. We first used the single-sided Fourier transform on the wideband filtered 0.5-12 Hz signal and computed phase derivatives as finite differences. We then numerically detected complex riding cycles which capture the generalized phase of the largest fluctuation on each pixel $\phi(x,y,t)$. By solving the corresponding Euler–Lagrange equations, phase velocity $\mathbf{v}_\varphi(x,y,t) = (\mathbf{u}(x,y,t),\mathbf{v}(x,y,t))$ can be calculated from the phases $\phi$[51]. Processing for the above methods can be found in the toolbox NeuroPatt [https://github.com/BrainDynamicsUSYD/NeuroPattToolbox][9].

### Order parameters of wave speed and direction

We introduced an order parameter $\bar{v}_\varphi$ to characterize the collective motion of the wave patterns:

$$\bar{v}_\varphi(t) = \frac{1}{Nv_0(t)} \left| \sum_{x,y} \mathbf{v}_\varphi(x,y,t) \right| \tag{1}$$

where $N$ is the number of vectors in the analysis window, $v_0$ is the average magnitude of the velocity over all pixels, and $\mathbf{v}_\varphi$ is the phase velocity $\mathbf{v}_\varphi(x,y,t)$. The homogeneity $\bar{v}_\varphi$ ranges from 0 to 1, with 1 representing the case where the velocity vectors are in parallel.

To quantify the heterogeneity of wave pattern dynamics, we introduced an index as in[52]:

$$H = \frac{1}{T} \sum_{t=1}^{T} \frac{1}{\mu(t)} \sqrt{\left( \frac{1}{N_t} \sum_{i=1}^{i=N} [v_i(t) - \mu(t)]^2 \right)} \tag{2}$$

where $T$ is the total time and $N_t$ is the number of vectors in the analysis window at time $t$, $v_i(t)$ is the speed (length) of phase velocity vector $i$ at time $t$, $\mu(t)$ is the mean vector speed at time $t$. If $H = 0$, all vectors have the same length, which indicates that the waves propagate at the same speed. A larger heterogeneity means a larger variance of the wave propagation speeds across space.

### Speed of large waves

We defined large waves as the positive period of the spatial average voltage oscillation, if the peak amplitude is higher than $1\times10^{-3}$ $\Delta R/R$. The voltage maps of large waves were additionally spatially filtered (2D Gaussian, $\sigma = 232$ $\mu$m). For each pixel recruited during a large wave, we computed the time of peak amplitude. For two neighboring pixels, the spatial distance divided by the peak time difference corresponds to the wave speed. Thus, for each map of peak time, we computed the local

spatial gradient for each pixel, and determined the propagation speed as the median of these gradients.

## Identification of wave patterns

We identified wave patterns based on previous methods[14] with some modifications. Plane waves were defined by periods when homogeneity $\bar{v_\varphi}(t) \geq 0.85$ within each cortical hemisphere inside the data acquisition field of view. Variation of the threshold values between 0.8 and 0.9 did not substantially change the results.

Standing waves were defined as periods with no apparent wave velocity (i.e. propagation) over an entire hemisphere within the data acquisition field of view. The criterion is that an average magnitude of the velocity fields is 2 SD below the mean value across the analyzed time period.

Organizing around the critical points, local complex wave patterns were identified by the intersections of two bilinearly interpolated null clines of the phase velocity field. Pattern types (source, sink or saddle) were further classified by the eigenvalues of the Jacobian matrix at the corners of the four pixels around the critical point[14,17]:

$$J = \begin{pmatrix} \frac{\partial u}{\partial x} & \frac{\partial u}{\partial y} \\ \frac{\partial v}{\partial x} & \frac{\partial v}{\partial y} \end{pmatrix} \tag{3}$$

Based on the trace ($\tau$) and determinant ($\triangle$) of the Jacobian matrix, potential source (unstable point, $\tau > 0$), sink (stable point, $\tau < 0$) and saddle ($\triangle < 0$) patterns were determined[14,17]. Two additional criteria must be satisfied for source and sink patterns[17]: 1) the angle between two nearby vectors should be less than $\alpha \bullet 2\pi/N_v$. $\alpha$ is a parameter to adjust the threshold, here we used $\alpha = 1.2$. $N_v$ is the number of vectors around a circle with a certain radius; 2) the angle between diagonal vectors should be less than $\beta \bullet 2\pi$, $\beta$ is also a parameter to adjust the threshold, here we used $\beta = 0.3$. We counted each unstable 'node' (both eigenvalues are real and of the same sign) or 'focus' (eigenvalues are complex-conjugate) as a source event, and stable 'node' or 'focus' as a sink event. That is, we do not distinguish 'node' from 'focus' when considering source and sink. We defined the center location of every source, sink and saddle pattern as the singularity point of PVF with near-zero velocity. We defined the duration of these complex waves as the number of time steps that the same location is occupied by the same singularity. For each pixel and for each wave pattern type, we calculated the probability of pattern emergence as the cumulative time that the pattern existed.

To validate our method for detecting sources, sinks and saddles (local waves), we compared the probability of local waves detected in the real data and the shuffled data as a function of the detection threshold. Shuffling was performed by randomization of the phase of the Fourier components of raw voltage signals, thus preserving the power spectrum on each pixel and random shuffling of pixel indices. This detection threshold was defined as a pair of values ($d, r$) where $d$ is the lifetime duration (number of time steps) of the same local wave pattern, and $r$ is the minimum radius (number of pixels from the singularity) of the local wave pattern. The probability at detection threshold ($d, r$) is defined as the number of local wave patterns with detection threshold ($d, r$) divided by the number of local wave patterns with detection threshold (1, 1). The detection threshold (2, 3) was chosen for the experimental analysis based on the 95% confidence that the detected waves are not due to randomness (Supplementary Fig. 1).

## Singular value decomposition of wave patterns

To identify the principal components of the wave patterns, we applied singular value decomposition to all phase velocity fields $\mathbf{v}_\varphi(x,y,t) = (\mathbf{u}(x,y,t), \mathbf{v}(x,y,t))$[9]. Every frame of the imaging data had a matrix of phase velocity field vectors, from which we derived a matrix $\mathbf{w}$ containing $(\mathbf{u}(x,y,t), \mathbf{v}(x,y,t))$. The singular value decomposition can

be defined as:

$$\mathbf{w} = \mathbf{T} \sum \mathbf{R}^* \tag{4}$$

where $\mathbf{T}$ and $\mathbf{R}$ are unitary matrices, * denotes the conjugate transpose, and $\sum$ is a diagonal matrix of the singular values $\sigma$. The $k$-th spatial mode, defined by the velocity field in the $k$-th column of $\mathbf{R}$, has a proportion of the overall variance given by $\sigma_k^2 / \sum \sigma_i^2$. Then we projected the instantaneous representative PVFs on the principal modes to obtain:

$$\mathbf{M} = \mathbf{w}/\mathbf{R} \tag{5}$$

where $\mathbf{M}$ is the weight matrix of every principal mode contributed to the PVFs. From this we can obtain the projection variance of the $m$-th spatial mode on the $n$-th PVFs that $M_{m,n}^2 / \sum_i M_{i,n}^2$.

## Neural mass model

The neural network model that we used for understanding the transition of wave patterns is a minor modification of the model proposed by Steyn-Ross et al.[53]. Unlike their original model, which only considers the effect of anesthesia on GABAa synaptic receptors, we modeled the effect of anesthesia on both AMPA receptors, GABAa receptors, and gap junctions, which is a more biologically realistic scenario. We modeled a cortical area of size $L^2 = 1\,cm^2$ (similar in size to the mouse cortical field of view of our imaging experiments) and composed of $N \times N = 100 \times 100$ lattices, where each lattice, with the size 100 $\mu m^2$ represented a tissue volume element (voxel) that can be regarded as a cortical column. For simplicity, we adopted a periodic boundary in the simulation. The excitatory (E) and inhibitory (I) membrane potentials at the column in position $(x,y)$ at time $t$, $V_e(x,y,t), V_i(x,y,t)$, obey the dynamic equations

$$\begin{cases} \tau_e \frac{dV_e(x,y,t)}{dt} = V_e^{rest} - V_e + (V_e^{rev} - V_e)g_e\Phi_e + (V_i^{rev} - V_e)g_i\Phi_i + D_e\nabla^2 V_e, \\ \tau_i \frac{dV_i(x,y,t)}{dt} = V_i^{rest} - V_i + (V_e^{rev} - V_i)g_e\Phi_e + (V_i^{rev} - V_i)g_i\Phi_i + D_i\nabla^2 V_i, \end{cases} \tag{6}$$

where the excitatory membrane time constant, the resting potential and the reversal potential are $\tau_e = 40\,ms$, $V_e^{rest} = -62.5\,mV$ and $V_e^{rev} = 0\,mV$, respectively. The inhibitory membrane time constant, the resting potential and the reversal potential are $\tau_i = 40\,ms$, $V_i^{rest} = -64\,mV$ and $V_i^{rev} = -70\,mV$, respectively. The E and I gap junction diffusion-coupling strengths[54] are $D_e = D_i \times 10^{-2}$ and $D_i = 0.07\,mm^2/ms$. The synaptic strengths are $g_e = 0.156$ and $g_i = 0.875$.

The synaptic conductance $\Phi_e, \Phi_i$ are rate-driven response with alpha function form. They obey the equations

$$\begin{cases} (\tau_d^E \frac{d}{dt} + 1)^2 \Phi_e(x,y,t) = N_e^{cc}\Phi_e(x,y,t) + N_e^{local}Q_e(V_e) + I^{sc} + a\sqrt{I^{sc}}\xi_e(x,y,t), \\ (\tau_d^I \frac{d}{dt} + 1)^2 \Phi_i(x,y,t) = N_i^{local}Q_i(V_i). \end{cases} \tag{7}$$

The E and I synaptic decay times are $\tau_d^E = 5\,ms$, $\tau_d^I = 20\,ms$. Both E and I conductances are driven by the local mean E and I firing rates $Q_e(V_e), Q_i(V_i)$, which are modeled by commonly used sigmoid functions

$$Q_b(V_b) = \frac{Q_b^{max}}{1 + \exp(\pi(\theta_b - V_b)/(\sqrt{3}\sigma_b))}, b = e,i \tag{8}$$

The parameters in the sigmoid functions are maximum firing rates $Q_e^{max} = 0.03/ms$, $Q_i^{max} = 0.06/ms$, threshold voltages $\theta_e = \theta_i = -58.5\,mV$ and standard deviations $\sigma_e = 3\,mV$, $\sigma_i = 5\,mV$.

Apart from the local source, excitatory conductance is also driven by corticocortical current flux $\phi_e$, modeled by the wave equation[55]

$$\left(\frac{r}{v}\frac{d}{dt}+1\right)^2\phi_e(x,y,t) = Q_e(V_e) + r^2\nabla^2\phi_e, \tag{9}$$

where the axonal conduction velocity is $v = 0.056\text{mm/ms}$ and the characteristic length scale for corticocortical axonal connectivity is $r = 0.1\text{mm}$. The axonal connection numbers of corticocortical E connection, local E connection and local I connection in Eq. (7) are set as $N_e^{cc} = 200$, $N_e^{local} = 85$, and $N_i^{local} = 120$ respectively. Finally, the noisy subcortical inputs (e.g. from thalamus) to excitatory conductance are also included and the strength is $I^{sc} = 0.018\text{/ms}$ with $a = 5.2$ being the scale factor of noise strength. $\{\xi_e(x,y,t)\}$ are modeled as independent Gaussian white noise with zero mean and unit variance. The above model represents the wakeful state.

In the model, we considered three effects of barbiturates on excitatory synaptic receptors (such as AMPA receptor), inhibitory receptors (such as GABAa receptor)[26] and gap junctions[25]. First, barbiturates can prolong the decay time of the inhibitory post-synaptic potential while maintaining the height of the peak[18]. Second, barbiturates can decrease the peak of the excitatory post-synaptic potential without affecting its decay time[19]. Third, barbiturates can block gap junction coupling[20]. Taking together, we can introduce three scale factors $\Delta\lambda_i \geq 0, \Delta\lambda_e \geq 0, \Delta\lambda_D \geq 0$ to rescale the parameters in the model such that

$$\begin{cases} \tau_d^I \to (1+\Delta\lambda_i p)\tau_d^I \\ g_i \to (1+\Delta\lambda_i p)g_i \end{cases}, g_e \to (1-\Delta\lambda_e p)g_e, \begin{cases} D_e \to (1-\Delta\lambda_D p)D_e \\ D_i \to (1-\Delta\lambda_D p)D_i \end{cases}, \tag{10}$$

which represent the anesthetic effects on the inhibitory synapse, excitatory synapse and gap junction respectively. We used a common modulating parameter $p$, where a larger parameter $p$ value indicates stronger anesthetic effect. Here, the parameters $\Delta\lambda_i = 0.08, \Delta\lambda_e = 0.01, \Delta\lambda_D = 0.4286$ reflect the relative difference of the effects[26,54]. To compare the results of recovery from anesthesia to wakefulness, we simulated the cases of 6 parameters where $p$ ranges between 0.5 and 0 in decreasing steps of 0.1, corresponding to states 1-6 in Fig. 6. Theoretical analysis of the dynamical states and parameter regions are shown in SI (Supplementary Fig. 5).

Long-range connections play an important role in the cortical network[30,56]. To further study the effect of long-range connections in different anesthetic states, an additional term $\phi_e^{LR}(x,y,t)$ is introduced into the right-hand-side of the first equation of Eq. (7) as follows,

$$\phi_e^{LR}(x,y,t) = \sum \mu Q_e(i_2,j_2,t-T_{1,2})\delta_{xi_1}\delta_{yj_1}. \tag{11}$$

Each term in the summation in Eq. (11) represents the effect of a long-range link from site $(i_2,j_2)$ to site $(i_1,j_1)$ with strength $\mu$. $\delta$ is the Kronecker delta function, such that $\delta_{xi_1}\delta_{yj_1} = 1$ if and only if $x = i_1, y = j_1$. $T_{1,2} = \sqrt{(i_1-i_2)^2+(j_1-j_2)^2}/v$ is the transmission delay[53], which is proportional to the distance between the two sites.

Numerical simulations were implemented by a forward difference method with a time step $dt = 0.4\text{ms}$ and the Laplacian operator in Eqs. (7) and (9) was computed by a five-points central formula. For simulating the dynamics without long-range connections, for each parameter setting, we simulated the model for 60 s for random initial condition. The simulation outcome for each set of parameters was obtained by averaging across 10 independent trials. For simulating the dynamics with long-range connections where we studied the regional signal coherence, we simulated the model for 500 s for random initial condition and the results were obtained from 100 independent trials, as the coherence measure showed strong variation across trials.

For the analysis of the modelling results, the voltage maps $V_e(x,y,t)$ were bandpass filtered (0.5-12 Hz). Similar to processing the voltage maps of the experimental data, we used the detection threshold (2, 3) to exclude local wave patterns which may represent non-organized activity.

## Coherence

We used Magnitude-Squared Coherence to estimate the correlation as a measure of similarity in the activity between two locations. The magnitude-squared coherence is a function of the power spectral densities, $P_{xx}(f)$ and $P_{yy}(f)$, and the cross power spectral density, $P_{xy}(f)$, of x and y:

$$C_{xy}(f) = \frac{|P_{xy}(f)|^2}{P_{xx}(f)P_{yy}(f)} \tag{12}$$

With the values of $C$ ranging between 0 and 1, indicating how well x corresponds to y at each frequency. A larger value of coherence reflects higher correlation between x and y.

## Statistical analysis

No statistical method was used to predetermine sample size. No data were excluded from the analyses. The experiments were not randomized. The Investigators were not blinded to allocation during experiments and outcome assessment. The name and type (one-sided or two-sided) of the statistical test, the N value for each statistical analysis, the comparisons of interest are stated together with the results. All applied tests are nonparametric and thus no assumption is required on the data distribution.

## Reporting summary

Further information on research design is available in the Nature Portfolio Reporting Summary linked to this article.

# Data availability

The experimental and simulation data generated in this study are available on Zenodo [https://doi.org/10.5281/zenodo.7574791].

# Code availability

All code for performing the analyses in this study are available on Github [https://github.com/MianxinLiu/Complexity-of-cortical-wave-patterns-of-the-mouse-cortex], and Zenodo [https://doi.org/10.5281/zenodo.7497560]. Code for phase velocity field calculations can be found in the toolbox NeuroPatt [https://github.com/BrainDynamicsUSYD/NeuroPattToolbox].

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

## Acknowledgements

This work was supported by the Hong Kong Research Grant Council (Grant HKBU12200217, HKBU12200620, HKBU12200421), and the National Science Foundation of China (Grant 11975194) to C.Z.; National Institutes of Health BRAIN Initiative Grants 1U01-MH-109091 and 5U01-NS-099573 to T.K.; and the Australian Research Council (Grant DP160104316) to P.G.

## Author contributions

T.K., C.Z. and P.G. designed research; C.S. and T.K. performed experiments; Y.L., M.L., P.G., C.Z. and T.K. analyzed data; J.L. and C.Z. built the model, J.L. and Y.L. simulated and analyzed the model; all authors engaged in writing the paper.

## Competing interests

The authors declare no competing interests.
