## [Peer Review File · Nature Communications]

Complexity of cortical wave patterns of the wake mouse cortexREVIEWER COMMENTS

Reviewer #1 (Remarks to the Author):

This is an interesting and very complete description of brain activity during wakefulness and unconsciousness from the perspective of wave phenomena. Overall, I think that the authors did an excellent job. The manuscript is engaging, figures are informative, also the methods are well described. I have only a few comments that I hope contribute to further improve the quality of this manuscript.

1. Looking at the spectral power for wakefulness and sleep in Figure 1, I see that during wakefulness, not only the low frequency peak is decreased, but also its position is displaced towards the left, i.e. even lower frequency. This, together with the observation of increased power in higher frequencies, suggests that the spectrum could be the same but "rotated", see: Donoghue, T., Haller, M., Peterson, E. J., Varma, P., Sebastian, P., Gao, R., ... & Voytek, B. (2020). Parameterizing neural power spectra into periodic and aperiodic components. *Nature neuroscience*, 23(12), 1655-1665.
2. Are the waves harmonic? Maybe it could be informative to plot the average waveform for each state at different frequency bands, to see if the shape of the waves being compared, in the time domain, are similar.
3. Computing the curl and the divergence could summarize in a straightforward way some properties of the vector fields. This could be helpful since the complete description of the wave patterns ends up being somewhat overwhelming (although, as I said before, very complete)
4. In panel F of Figure 1 I understand that the background color indicates the level of activity at that particular voxel, so we see states with overall high activity and others with overall lower activity. What is the relationship between the properties of the wave propagation vector field and the level of activity?
5. Voltage imaging has been used before to study unconsciousness induced by anesthesia in this paper: Scott, G., Fagerholm, E. D., Mutoh, H., Leech, R., Sharp, D. J., Shew, W. L., & Knöpfel, T. (2014). Voltage imaging of waking mouse cortex reveals emergence of critical neuronal dynamics. *Journal of Neuroscience*, 34(50), 16611-16620. However, the authors of this paper employed a different perspective, characterizing dynamics from the viewpoint of critical phenomena, in particular, analyzing the properties of avalanche distributions. I wonder what do you think about how your results relate to theirs, and whether you might be describing the same phenomena from different angles
6. When introducing your model, you wrote: "We modeled the effect of barbiturate anesthesia on AMPA and GABA_A receptors and gap junctions (Rudolph and Antkowiak, 2004; Johansson, 2006; Jin et al., 2010), with an overall effect that can be described as a reduction of local circuit excitability". Is this correct? Barbiturates act as GABA agonists, hence I would model their effects as increased inhibition instead of decreased excitation.
7. Most of your results collapse the temporal dimension, however, the fields were obtained at each time point. Is there something interesting to say about how (un)consciousness modulates the dynamics of the field properties?

Reviewer #2 (Remarks to the Author):

Increased complexity of cortical wave patterns of the conscious brain

The authors studied the evolution of cortical wave patterns using optical voltage imaging in mice transitioning from barbiturate-induced anesthesia to wakefulness. Using phase velocity field analysis, they observed that there was a reduction in global voltage waves and an increase in local wave events and complexity as the brain transitioned into wakefulness. A neural mass model was employed which incorporates barbiturate-action on chemical synapse kinetics and coupling strength of gap junctions. This model could show how competition between global and local patterns and long range connections can recapitulate empirical observations.

Overall this is quite an interesting and relevant study, combining experimental and advanced modelling approaches in rodents to assess cortical wave dynamics as a level of awakesness.

I do not have many major issues with the paper. However, I would suggest to add further statistical testing results / data in order to support the description of the findings. Moreover, a sample size of 5 mice seems quite small for the experimental portion, so one would ideally want to see whether findings replicate in either a larger or independent sample of mice.

Specific comments are seen below:

1) P4.I.4 (Fig 1A) Please provide statistical testing results to support reductions in low frequency power and increases in high frequency power. Please provide standard deviations in Fig 1 around anes and awake lines. Fig 1B Please consider using a different colormap than yet.

2) P5. line 7, Fig 2C - please provide statistical results of the comparison together with error bars for each of the mice.

3) The authors run a PCA of phase velocity fields to evaluate spatiotemporal features of activity of the waves. Please clarify whether the modes were obtained from group average or single subject data. if the former, are patterns consistent at the single subject level (eg after components were aligned using procrustes rotations)?

In order to compare states, wouldn't one ideally also want to build a 'template' PCA space that is derived from both states and then see how the scores in both states in this space differ? At the moment, it is not fully clear to me whether the decompositions between states can be compared that easily, as higher order modes (eg mode 5) do not appear to be that comparable between states. Perhaps the authors want to briefly comment on this, and/or present these additional analyses in supplementary findings.

4) Figure 7A and B would also benefit from error bars around the line plots, in both model and simulation.

5) P.21 - please provide empirical support/references for why anesthesia is more likely to affect AMPA, GABA_A, and gap junctions. Also, based on prior results, wouldn't one expect these effects to be heterogenous across the brain, based on regional variations in eg GABA_A receptor density? A few comments on that in the discussion could be worthwhile.

Reviewer #3 (Remarks to the Author):

In this work, the authors consider differences in brain dynamics during the transition from anesthetized to the awake state. They use computational analysis techniques developed previously to understand how dynamics across the cortex change during recovery from barbiturate-induced anesthesia to the awake state. They then use a neural mass model to illustrate a possible mechanism for how cortical dynamics change during this transition.

The analysis techniques and results are well-executed and presented. The model is somewhat general, and while it may not capture the precise network mechanisms for the changes in dynamics, it illustrates well why one might expect to find these changes in large-scale cortical dynamics following changes in brain state.

Overall, the paper is well-done, solid, and interesting. The results shed light on one of the most interesting questions in systems neuroscience, showing how different forms of activity can arise from the same circuit in different dynamical regimes.

Below are some comments that should be addressed in this submission.

MAJOR

- One major methodological concern is with the definition of "wakeful state" (as on page 3, lines 35-36) as "indicated by occasional spontaneous coordinated body movements". From the perspective of an outsider to veterinary anesthesia, this definition of wakefulness seems potentially problematic, because it is not at all clear whether "occasional spontaneous coordinated body movements" is consistent with normal waking physiology. If there is some question here, it may be important to compare the results in this present manuscript with the same imaging from animals that have not been anesthetized, to ensure that the cortical dynamics following recovery from anesthesia are consistent with those from the physiologically normal state in awake mice.

- Fig. 2B,C: The results on propagation speed reported here differ from previous work in imaging in the rat (see Xu et al., Neuron, 2007; Han et al., Neuron, 2008 among others), which reported waves propagating from 10-70 mm/s. Given that even single cortical regions in the mouse can extend over several millimeters, the average wave speeds of 1-3 mm/s are somewhat difficult to understand. Consider, for example, that mouse V1 is approximately 4 mm across. Does this mean that the waves observed here, whose temporal frequency ranges from 0.5-12 Hz, would take 3-4 seconds to propagate over that cortical area? Could these results be affected by smoothing in the phase analysis?

- Fig. 6A: Do the wavelengths differ between the anesthetized and awake states in the model?

- Fig. 6B,C,D: Units should be specified here.

- page 18, line 38: What is "generalized phase"?

MINOR

- Fig. 1D: There should be a scale bar along the spatial dimension of this figure.

- page 12, line 19: What does "inspired" indicate here?

- page 21, line 10: The authors should specify in more detail how the phase velocity field is represented using complex numbers.

Reviewer #4 (Remarks to the Author):

In their study, the authors show a change in the characteristics of ongoing propagating waves of activity across the neocortex during transition from the anesthetized state towards wakefulness. In comparison to the anesthetized state, wave motifs in wakefulness remain overall more local and display a higher degree of complexity with decreased speeds. Higher speeds of waves are, however, observed in fewer cases of large wave events during wakefulness. Furthermore, the authors present a neural mass model supposed to capture the main characteristics of the results.

The results were obtained using genetically encoded voltage indicator (i.e., expression of fluorescent proteins bound to a voltage-sensing membrane protein in pyramidal neurons) in combination with wide-field optical imaging in mice.

To my opinion the study deals with an important topic in brain science using advanced recording techniques and analysis methods. What I find the main disturbing factor, however, is a potential overselling of the drawn conclusions with respect to the underlying evidence/hypotheses and partially a severe lack of statistical presentation of the results, as detailed below.

In fact, I think a sound presentation (including statistical evidence) and discussion of neocortex-wide dynamics of voltage changes after removal of anesthesia in the given spatial and temporal resolution over the mouse brain – nowadays the most intensively used animal model to investigate brain activity worldwide – could (have been) in principle be of sufficient interest for the readership of Nature Communications.

Detailed comments:

Overall it appears not too surprising that cortical dynamics become more complex during recovery from anesthetics. The authors claim in their title “Increased complexity of cortical wave patterns of the conscious brain”. In how far were the animals conscious? The authors wrote in this context: “... mice reached a wakeful state as indicated by occasional spontaneous coordinated body movements.” To my opinion such behavioral observations are not the best predictor of the state of the animal. Could the authors provide some more objective measures of the state of the animal during recording?

An adjective like “conscious” is at least debatable in the context of its use for animals. I have principally no strong objections against using it for animals given some discussion about its definition. Anyways, in view of the above mentioned limitations in characterization and quantification of the animals’ state and to avoid that the reader might shape wrong expectations the used species should at least be mentioned in the title and abstract.

The authors are widely using the term “wakeful” to describe the recorded cortical state after anesthesia. In contrast, the term “awake” is consistently used in the figures and legends, and occasionally in the main text. I am not a native speaker; however, I think that the latter term would again need even stronger justification.

Movement artifacts and assignment of spontaneous activity: In the anaesthetized state epochs with movement artefacts are naturally lower (or ideally non-existent) compared to the awake state. This factor should be considered for hypothesis testing (what was the remaining percentage of employed trials or time epochs after the exclusion criterion was applied?). An alternative criterion to select epochs with ongoing activity might be to correlate movements of the animals (whisking, grooming, moving, eye movements, etc.) with the recorded frames. This might further justify exclusion of those time epochs that are not considered as representing ongoing activity. Some of these behaviors (and brain activity thereafter) might not produce movement artefacts but pass the selection threshold of the currently applied “amplitude > 3 standard deviation”. As a consequence, the higher occurrence of local waves in the awake state may be related to such behaviors and therefore display evoked contributions rather than reflecting ongoing activity.

Given the relatively weak objective criteria that the authors seem to have at hand to state that the recordings monitored conscious processing (animals were also not engaged in a task), the extensive discussion about the relationship between their results and conscious processing seem to my opinion beyond what is covered by their data. Instead, I suggest adding more detailed and critical discussion about possible limits of their study. For example, how can the recorded signals be interpreted, do they majorly reflect post- or presynaptic activity? Do the observed wave trajectories strictly match/follow anatomical cortical connection patterns? Could wave events be shadowed by different amounts of contributions from pre- and postsynaptic events?

Do the authors believe that the observed waves are strictly generated within the neocortex? Unless I overlooked it, the model seems to incorporate solely intra-cortical interactions? Thus, do the authors exclude contributions from subcortical regions?

In how far is the model scalable to larger (e.g., human) brain sizes?

Line 22, page 2: It should be made clear which references are related to traveling waves found in spontaneous activity patterns and which references refer to stimulus-triggered waves (e.g., Benucci et al. 2007 and Muller et al., 2018). Moreover, Jancke et al., 2004 should be added here, as this study was the first demonstration of subthreshold propagation vs. suprathreshold propagation of population activity using voltage indicator imaging in direct combination with extracellular recordings. As a side remark, please note that the claim of Benucci et al. 2007 to have distinguished traveling waves from a standing wave was called into question by Sit et al., 2009.

Given the extensive statements about the speed of waves, speed values should be presented as numbers including statistics. Unless I overlooked it, values are displayed only (in Fig. 2).

Several figures represent a single experiment/animal without reporting values or performing statistical tests. In this way, the authors' statements heavily rely on the visual inspection of the figures by the reader. This is a major drawback in my opinion and hinders many of the conclusions drawn from the data. In addition, it should become clear in the main text (rather than in figure legends only) whenever single representative examples are described. E.g., line 1-11, page 4: "In both the anesthetized and wakeful states, we observed a maximum frequency power of the cortex-wide population voltage activity ..."

Here, a list of some further examples of statements with missing statistics:

Lines 2-9, page 5: "... we first quantified the propagation direction and speed of the detected waves. The voltage waves propagated in preferred directions at both brain states, but the wave directions are more restricted to the frontal to parietal direction in the anaesthetized state"

"The wave propagation speeds showed a broader distribution in the wakeful state (Fig. 2B)."

"The average speed of PVF decreased (Fig. 2C) even though the average speed of large waves increased (Fig. 2C inset) from anesthetized to wakeful states for all mice analyzed (N = 5 mice)."

Lines 6-28, page 6: none of the claims are supported by statistics.

Line 27, page 6. "This revealed that the top 5 modes are similar between the anaesthetised and wakeful states (Fig. 3C)." – According to this figure, we see instead how much the SVD modes are similar to the reference mode (1st mode of the anaesthetized state). A straightforward correlation analysis between these modes in both states might justify the above-mentioned statement.

Line 28, page 6: "Results are similar for the other four mice" – Where and how is this statement supported?

Finally, the connection of the model to the empirical data appears somewhat weak. E.g.:

Lines 13-14, page 13 “As the strengths of the long-range connections increase, the coherence under the awake state increased whereas the coherence under the anaesthetised state does not show an increase (Fig. 7C).” – However, empirical data in Fig. 7D seem to a considerable extent not to match the simulation shown in Fig. 7C?

Also depictions in Fig. 6G (model) and Fig. 5C (data) “Probability of local complex wave number (source, sink and saddle) ...” do not show a convincing similarity.

Minor

Line 9, page 2: “More recently, a large body of evidence supported the idea that the increasing complexity of cortical activities is a signature of recovery from general anesthesia to wakefulness or conscious states ...”

Does this statement make sense? To me it reads as if an “increasing complexity of cortical activities” is strictly/exclusively bound to such specific transition. Maybe omitting “the” (before “increasing”) helps? Alternatively: More recently, a large body of evidence supported the idea that a signature of recovery from general anesthesia to wakefulness or conscious states is an increasing complexity of cortical spontaneous activities?

Throughout the text the authors state that “long-range connections” play a major role in the generation of more complex pattern dynamics observed in wakeful states. In the literature (particularly when dealing with larger brain sizes) long-range connections are often associated with horizontal connections within a cortical area. Do the authors refer to “long-range” spanned exclusively between different cortical areas? How were the dimensions of “long-range” specified here?

Related to Fig. 1: “We observed spatiotemporal patterns in the color-coded voltage maps (Fig. 1C, D), with activity waves that appeared more widespread during the anesthetized state (Fig. 1E).” – During transition to wakefulness, there seems a reduction of wave amplitudes? Is this a true effect? Otherwise, it may be helpful to color-scale plots in Fig. 1D individually.

Propagation distance of the waves appears neither quantified in the empirical data nor in the simulated one.

The authors categorize the animals’ state as either wakeful or anesthetized. According to the literature, recovery from pentobarbiturate is quite variable and can take relatively long (e.g., Field et al., 1993, Haberham et al., 1998). What were the concrete times after the bolus injection during which the recorded trials were considered as reflecting wakefulness?

Furthermore, could the authors find gradual changes (as opposed to only a binary characterization wakeful or anesthetized) of average speed, heterogeneity etc. in their data - similar as stated in their model (Fig. 6)?

In Fig. 6 annotation of speed unity on y-axis is missing.

Line 37, page 16 – line 3, page 17: “... but we shall note that in the real cortex and under physiological conditions, increased excitability during the conscious state are likely to be mediated by endogenous neurostimulators such as acetylcholine and amines.”

The authors might substantiate their proposition with a citation: In a recent review Jancke and colleagues demonstrated changes of ongoing brain dynamics based on activation of a single amine receptor type (5-HT_{2A}) and outlined experimental options for such manipulation (Jancke et al., 2021). In addition, the authors show its different effects on spontaneous activity under anesthesia and awake conditions (Azimi et al., 2021).

REVIEWER COMMENTS

Original text by reviewers is copied in black font; our responses are marked by **green bold font**.

Reviewer #1 (Remarks to the Author):

This is an interesting and very complete description of brain activity during wakefulness and unconsciousness from the perspective of wave phenomena. Overall, I think that the authors did an excellent job. The manuscript is engaging, figures are informative, also the methods are well described. I have only a few comments that I hope contribute to further improve the quality of this manuscript.

We very much appreciate the kind and encouraging words from the reviewer – thank you.

1. Looking at the spectral power for wakefulness and sleep in Figure 1, I see that during wakefulness, not only the low frequency peak is decreased, but also its position is displaced towards the left, i.e. even lower frequency. This, together with the observation of increased power in higher frequencies, suggests that the spectrum could be the same but "rotated", see: Donoghue, T., Haller, M., Peterson, E. J., Varma, P., Sebastian, P., Gao, R., ... & Voytek, B. (2020). Parameterizing neural power spectra into periodic and aperiodic components. *Nature neuroscience*, 23(12), 1655-1665.

In our study, we have focused on broadband activity (0.5-12Hz) as in (Davis, et al. 2020), so the possible confounding effect of aperiodic activity on narrow-band oscillations would not be of relevant concern. Nevertheless, in our revised manuscript, we have cited the reference highlighted by the reviewer and pointed out the changes of neural PSD are also state-dependent as nicely shown in Donoghue et al. 2020 (P.4).

2. Are the waves harmonic? Maybe it could be informative to plot the average waveform for each state at different frequency bands, to see if the shape of the waves being compared, in the time domain, are similar.

The waves are not periodic and have a non-sinusoidal waveform. We showed the frequency spectrum essentially to link with classical descriptions of cortical population activity and to justify our statement that the frequency band chosen captures most of the above 1/f background-power. In direct response to the reviewer's specific suggestion, we now show in Fig. R1 below the average waveforms before and after the application of our band-pass filter; they do show some differences for the different brain states.

Fig. R1: Average wave form from the up-down oscillations aligned at the zero point. Left: averaged waveforms before bandpass. Right: averaged waveforms after 0.5-12 Hz bandpass.

3. Computing the curl and the divergence could summarize in a straightforward way some properties of the vector fields. This could be helpful since the complete description of the wave patterns ends up being somewhat overwhelming (although, as I said before, very complete)

Thank you for this suggestion. We have now calculated the curl and the divergence based on the PVF; these new results below (Fig. R2) have been added to Supplementary information as Fig. S2. The areas with higher curl and the divergence values tend to have more spiral waves and sources/sinks, respectively. The relevant text summarizing this has been added on P. 7 of the main text.

Fig. R2: Curl and the divergence to show properties of the vector fields. (a) Phase velocity fields (PVFs) with singularities. Background color is the voltage amplitude. (b) Curl of the PVFs in (a). (c) Divergence of the PVFs in (a).

4. In panel F of Figure 1 I understand that the background color indicates the level of activity at that particular voxel, so we see states with overall high activity and others with overall lower activity. What is the relationship between the properties of the wave propagation vector field and the level of activity?

The reviewer made an interesting point. Indeed, our PVF analysis quantifies the spatiotemporal alignment of phases of activities and is not directly related to their amplitude. In our original description of the methods (Townsend and Gong, 2018), following careful investigations we (Townsend and Gong) noted that the amplitude and

phase wave patterns are not always consistent with each other. In neural recordings, amplitude and phase data at the same frequency reflect different properties of brain activity, with amplitude representing a combination of the coherence and overall activity of a local neural ensemble and phase representing the timing of its oscillations. Accordingly, these signals typically contain different spatiotemporal patterns, and both phase and amplitude patterns can be relevant and informative. For this reason, we showed both amplitude and PVF in superposition in Fig 1 of this manuscript. We focused on the PVF as this allowed us to robustly detect relative spatiotemporal timing relationships of brain activity.

5. Voltage imaging has been used before to study unconsciousness induced by anesthesia in this paper: Scott, G., Fagerholm, E. D., Mutoh, H., Leech, R., Sharp, D. J., Shew, W. L., & Knöpfel, T. (2014). Voltage imaging of waking mouse cortex reveals emergence of critical neuronal dynamics. *Journal of Neuroscience*, 34(50), 16611-16620. However, the authors of this paper employed a different perspective, characterizing dynamics from the viewpoint of critical phenomena, in particular, analyzing the properties of avalanche distributions. I wonder what do you think about how your results relate to theirs, and whether you might be describing the same phenomena from different angles.

Thank you for raising this interesting point. Indeed, increased complexity is consistent with the reported approach of the critical state from the supercritical dynamics seen under light anaesthesia. In Scott et al. the cascaded process of activity amplitude above a threshold was analysed as avalanches. The waves detected by phase-based analysis in the current study not only covered the avalanches analysed in the Scott et al. paper, but also included the spatiotemporal patterns below the threshold, and focus on the continuous propagation expressed by phases. Chen and Gong 2019 highlighted a need to further study spatiotemporal organization of critical brain dynamics. From this perspective, some potential unification of wave and criticality could be achieved but this is beyond the scope of the present paper.

6. When introducing your model, you wrote: "We modeled the effect of barbiturate anesthesia on AMPA and GABA_A receptors and gap junctions (Rudolph and Antkowiak, 2004; Johansson, 2006; Jin et al., 2010), with an overall effect that can be described as a reduction of local circuit excitability". Is this correct? Barbiturates act as GABA agonists, hence I would model their effects as increased inhibition instead of decreased excitation.

Barbiturate anaesthesia was indeed modelled as increased inhibition (longer time constant) together with reduced excitability and transmission (Eq. 10), but this increased inhibition lowers overall excitability of the circuit (e.g. more difficult to be excited by excitatory synaptic drive). In other words, the same excitatory synaptic input produces less excitation (that is reduced excitability). In the revised manuscript, we have rephrased to avoid potential confusion.

7. Most of your results collapse the temporal dimension, however, the fields were obtained at each time point. Is there something interesting to say about how (un)consciousness modulates the dynamics of the field properties?

The key advance of our analysis is that we have focused on both the spatial and temporal properties of brain activity, going beyond traditional correlation-based analyses. The principal modes (fields) extracted from the PVF provide crucial spatial as well as temporal characterizations; these modes indeed change as the brain transitions into wakefulness. In the revised manuscript, we have further highlighted the differences of these modes and their changes during different brain states (P. 6-7).

Reviewer #2 (Remarks to the Author):

Increased complexity of cortical wave patterns of the conscious brain

The authors studied the evolution of cortical wave patterns using optical voltage imaging in mice transitioning from barbiturate-induced anesthesia to wakefulness. Using phase velocity field analysis, they observed that there was a reduction in global voltage waves and an increase in local wave events and complexity as the brain transitioned into wakefulness. A neural mass model was employed which incorporates barbiturate-action on chemical synapse kinetics and coupling strength of gap junctions. This model could show how competition between global and local patterns and long range connections can recapitulate empirical observations.

Overall this is quite an interesting and relevant study, combining experimental and advanced modelling approaches in rodents to assess cortical wave dynamics as a level of awakesness.

Thank you for the encouraging appraisal of our manuscript.

I do not have many major issues with the paper. However, I would suggest to add further statistical testing results / data in order to support the description of the findings. Moreover, a sample size of 5 mice seems quite small for the experimental portion, so one would ideally want to see whether findings replicate in either a larger or independent sample of mice.

We have now added statistical test results and analyses for more mice (see the below response).

Specific comments are seen below:

1) P4.l.4 (Fig 1A) Please provide statistical testing results to support reductions in low frequency power and increases in high frequency power. Please provide standard deviations in Fig 1 around anes and awake lines. Fig 1B Please consider using a different colormap than yet.

We have updated Fig.1 with results from fully wakeful state and have added SDs to Fig.1 and redrawn Fig.1b by using a different colour scheme.

2) P5. line 7, Fig 2C - please provide statistical results of the comparison together with error bars for each of the mice.

We have added these results to Fig. 2c-e.

3) The authors run a PCA of phase velocity fields to evaluate spatiotemporal features of activity of the waves. Please clarify whether the modes were obtained from group average or single subject data. If the former, are patterns consistent at the single subject level (eg after components were aligned using procrustes rotations)?

In order to compare states, wouldn't one ideally also want to build a 'template' PCA space that is derived from both states and then see how the scores in both states in this space differ? At the moment, it is not fully clear to me whether the decompositions between states can be compared that easily, as higher order modes (eg mode 5) do not appear to be that comparable between states. Perhaps the authors want to briefly comment on this, and/or present these additional analyses in supplementary findings.

Thank you for the suggestion. We have added new analyses using 'template' PCA space of all data trials and mice and check the variance difference in all three states of anesthetised, post woken and fully awake (Fig. 3). The descriptions are revised accordingly (P. 6-7).

4) Figure 7A and B would also benefit from error bars around the line plots, in both model and simulation.

Thank you for this suggestion. We have added error bars to Fig. 7a and b.

5) P.21 - please provide empirical support/references for why anesthesia is more likely to affect AMPA, GABA_A, and gap junctions. Also, based on prior results, wouldn't one expect these effects to be heterogeneous across the brain, based on regional variations in eg GABA_A receptor density? A few comments on that in the discussion could be worthwhile.

Our model of anaesthesia is overtly minimalistic aiming at some biological plausibility (i.e. increased inhibition, reduced excitability and transmission, see Eq. (10)) but without implying biological realism, to provide some mechanistic understanding of the key spatiotemporal dynamical features in the data. The essential effect is reduced circuit excitability (drive required to generate a wave, similar to excitability at the single neuron level defined as drive required to trigger an action potential). Most, if not all, agents that induce analgesia do so by mixed molecular mechanisms with the effect of reducing the excitation/inhibition ratio by acting on excitatory or inhibitory mechanisms or on both. We have cited Rudolph and Antkowiak, 2004; Johansson 2006 and Jin et al. 2010 papers as empirical evidence supporting our modelling assumption of the effect of anaesthesia. This effect is indeed heterogeneous across the brain but not replicated in our minimalistic model.

Reviewer #3 (Remarks to the Author):

In this work, the authors consider differences in brain dynamics during the transition from anesthetized to the awake state. They use computational analysis techniques developed previously to understand how dynamics across the cortex change during recovery from

barbiturate-induced anesthesia to the awake state. They then use a neural mass model to illustrate a possible mechanism for how cortical dynamics change during this transition.

The analysis techniques and results are well-executed and presented. The model is somewhat general, and while it may not capture the precise network mechanisms for the changes in dynamics, it illustrates well why one might expect to find these changes in large-scale cortical dynamics following changes in brain state.

Overall, the paper is well-done, solid, and interesting. The results shed light on one of the most interesting questions in systems neuroscience, showing how different forms of activity can arise from the same circuit in different dynamical regimes.

We thank the reviewer for the encouraging support of our manuscript.

Below are some comments that should be addressed in this submission.

MAJOR

One major methodological concern is with the definition of "wakeful state" (as on page 3, lines 35-36) as "indicated by occasional spontaneous coordinated body movements". From the perspective of an outsider to veterinary anesthesia, this definition of wakefulness seems potentially problematic, because it is not at all clear whether "occasional spontaneous coordinated body movements" is consistent with normal waking physiology. If there is some question here, it may be important to compare the results in this present manuscript with the same imaging from animals that have not been anesthetized, to ensure that the cortical dynamics following recovery from anesthesia are consistent with those from the physiologically normal state in awake

The reviewer made an important point. We have now included the analysis of data from mice that have not been anesthetized during imaging (and have not been exposed to any anaesthesia for more than 3 days prior to the recording sessions). We note that anaesthesia/sedation was required for the head implant and habituation procedure but well-habituated animals are tolerant to head fixation in the imaging setup while fully awake. We now show and describe these data explicitly, including the results of motion energy (ME) from facial video recordings synchronised to brain imaging, and heart rates to quantify the behavioural states (see Fig. R3 below). Analysis at several time periods during recovery from anaesthesia did reveal a smooth transition with respect to the wave features analysed in this study. These results are now presented in Fig. S3. This also addresses the similar comment of Reviewer #4, below.

Fig. R3: A smooth transition with respect to the wave features at several time periods during recovery from anaesthesia (a), in comparison with fully wakeful state (b).

- Fig. 2B,C: The results on propagation speed reported here differ from previous work in imaging in the rat (see Xu et al., Neuron, 2007; Han et al., Neuron, 2008 among others), which reported waves propagating from 10-70 mm/s. Given that even single cortical regions in the mouse can extend over several millimeters, the average wave speeds of 1-3 mm/s are somewhat difficult to understand. Consider, for example, that mouse V1 is approximately 4 mm across. Does this mean that the waves observed here, whose temporal frequency ranges from 0.5-12 Hz, would take 3-4 seconds to propagate over that cortical area? Could these results be affected by smoothing in the phase analysis?

For the coherent global planar waves that can sweep through larger regions, the speeds are large (Fig. 2c insert, 20-40 mm/s, ~0.1-0.2 second to propagate over V1 area) and the speed of such large wave events is larger during the wakeful state than during the anaesthetised state (Fig. 2c inset, Fig. 6b inset (model)). Note that the speeds of these large phase wave are consistent with those found in other existing studies (Rubino, Robbins, and Hatsopoulos, Nature Neuroscience, 2006). However, we found that those localised complex wave patterns including spiral and saddles have much slower speed (distribution peaked around 3 mm/s); the slow movements of these local waves happen because they possess phase singularities with zero speed and their propagations are often constrained by other surrounding patterns. Given our observations, we suggest that future studies should measure propagation speeds of all types of waves, rather than only focusing on big/global waves.

- Fig. 6A: Do the wavelengths differ between the anesthetized and awake states in the model?

We have now calculated the wavelengths using spatial Fourier transform on the simulated wave patterns from the model. The average wavelengths become smaller in the awake state compared to the anaesthetised state, consistent with the observation of more complex wave patterns in the awake state in Fig. 6a. The detailed description and results are added to SI (Fig. S4, S6) and briefly referred to in the main text.

- Fig. 6B,C,D: Units should be specified here.

We have added units in Fig. 6b. Fig. 6c is heterogeneity of the wave speeds and Fig. 6d is homogeneity of the wave directions which are order parameters range from 0 to 1.

- page 18, line 38: What is "generalized phase"?

The generalized phase is a method to extract appropriate phase to avoid phase distortions on narrow band filter, developed by Davis (Davis et al., 2020). We first used the single-sided Fourier transform on the wideband filtered 0.5-12Hz signal and computed phase derivatives as finite differences. Then, we numerically detect complex riding cycles which captures the generalized phase of the largest fluctuation on each pixel $\phi(x, y, t)$. We have now added this description to Methods.

MINOR

- Fig. 1D: There should be a scale bar along the spatial dimension of this figure.

A scale bar has been added, thank you.

- page 12, line 19: What does "inspired" indicate here?

We have now changed "inspired" to "suggested".

- page 21, line 10: The authors should specify in more detail how the phase velocity field is represented using complex numbers.

We have revised the description to clarify the issue: “we derived a matrix w containing $(u(x, y, t), v(x, y, t))$ ”.

Reviewer #4 (Remarks to the Author):

In their study, the authors show a change in the characteristics of ongoing propagating waves of activity across the neocortex during transition from the anesthetized state towards wakefulness. In comparison to the anesthetized state, wave motifs in wakefulness remain overall more local and display a higher degree of complexity with decreased speeds. Higher speeds of waves are, however, observed in fewer cases of large wave events during wakefulness. Furthermore, the authors present a neural mass model supposed to capture the main characteristics of the results.

The results were obtained using genetically encoded voltage indicator (i.e., expression of fluorescent proteins bound to a voltage-sensing membrane protein in pyramidal neurons) in combination with wide-field optical imaging in mice.

To my opinion the study deals with an important topic in brain science using advanced recording techniques and analysis methods. What I find the main disturbing factor, however, is a potential overselling of the drawn conclusions with respect to the underlying evidence/hypotheses and partially a severe lack of statistical presentation of the results, as detailed below.

In fact, I think a sound presentation (including statistical evidence) and discussion of neocortex-wide dynamics of voltage changes after removal of anesthesia in the given spatial and temporal resolution over the mouse brain – nowadays the most intensively used animal model to investigate brain activity worldwide – could (have been) in principle be of sufficient interest for the readership of Nature Communications.

We thank the reviewer for the stringent appraisal of our manuscript while supportively highlighting the importance of our work. We strived to have now addressed the reviewer’s concerns with our revision and agree that the reviewer’s comments further improved the quality of our manuscript; thank you.

Detailed comments:

Overall it appears not too surprising that cortical dynamics become more complex during recovery from anesthetics. The authors claim in their title “Increased complexity of cortical wave patterns of the conscious brain”. In how far were the animals conscious? The authors wrote in this context: “... mice reached a wakeful state as indicated by occasional spontaneous coordinated body movements.” To my opinion such behavioral observations are not the best predictor of the state of the animal. Could the authors provide some more objective measures of the state of the animal during recording?

The state of the animal was video monitored and, in our original submission, we described the observations as “no movement” and “occasional voluntary movements”. We agree that this subjective analysis may not be satisfactory and now extended our

analysis and documentation by qualifying the movement energy (ME) derived from the videos. We added the time course of ME along with representative monitoring video examples to Fig. S3 (see Fig. R2 in the above).

An adjective like “conscious” is at least debatable in the context of its use for animals. I have principally no strong objections against using it for animals given some discussion about its definition. Anyways, in view of the above mentioned limitations in characterization and quantification of the animals’ state and to avoid that the reader might shape wrong expectations the used species should at least be mentioned in the title and abstract.

The species is now stated both in the title and the abstract.

The authors are widely using the term “wakeful” to describe the recorded cortical state after anesthesia. In contrast, the term “awake” is consistently used in the figures and legends, and occasionally in the main text. I am not a native speaker; however, I think that the latter term would again need even stronger justification.

We clarified our wording and expanded our analysis to data obtained from mice in the fully awake state (anaesthetic-free). In one set of experiments, imaging began as the mice underwent light anaesthesia induced by a bolus injection of pentobarbiturate and continued until the mice woke up as indicated by occasional spontaneous coordinated body movements. We labelled the latter condition as “post woken”. Another data set was obtained from mice that were well-habituated to the imaging conditions and that had been free of anaesthesia for at least 3 days prior to the imaging session. We labelled this condition (brain state) as “fully awake”.

Movement artifacts and assignment of spontaneous activity: In the anaesthetized state epochs with movement artefacts are naturally lower (or ideally non-existent) compared to the awake state. This factor should be considered for hypothesis testing (what was the remaining percentage of employed trials or time epochs after the exclusion criterion was applied?). An alternative criterion to select epochs with ongoing activity might be to correlate movements of the animals (whisking, grooming, moving, eye movements, etc.) with the recorded frames. This might further justify exclusion of those time epochs that are not considered as representing ongoing activity. Some of these behaviors (and brain activity thereafter) might not produce movement artefacts but pass the selection threshold of the currently applied “amplitude > 3 standard deviation”. As a consequence, the higher occurrence of local waves in the awake state may be related to such behaviours and therefore display evoked contributions rather than reflecting ongoing activity.

In order to avoid the complications outlined by the reviewer, we did not exclude episodes based on the “amplitude > 3 standard deviation” criterion. If movement artefacts were suspected, we routinely excluded the whole seconds-lasting trial. We checked our records and in the sessions (1 session per day per animal) analysed, no trial were excluded. For clarity, the following statement in the first paragraph of the Results section was removed in the revised manuscript: “Periods of large body movements were excluded from analysis so that only activity during the quite resting state was analysed”.

Analysis of the frame-synchronized facial and cortical voltage images (see new data and results) confirmed that facial movements as well as minor movements of the forelimbs did not produce movement artefacts in the optical signals. For this reason, we could extend our analysis to fully awake mice which exhibited minor motor activity (while still accepting the habituated head restraint). Of course, ongoing activity may

relate to ongoing behaviours, some of which can be captured by facial expressions and other motor output (see e.g. doi: 10.1038/s41593-019-0502-4.). We do not know what, and if, a mouse is “thinking” but it is clear that there is ongoing behaviour associated with ongoing activity. Hence, our results and conclusions are not dependent on these considerations.

Given the relatively weak objective criteria that the authors seem to have at hand to state that the recordings monitored conscious processing (animals were also not engaged in a task), the extensive discussion about the relationship between their results and conscious processing seem to my opinion beyond what is covered by their data. Instead, I suggest adding more detailed and critical discussion about possible limits of their study. For example, how can the recorded signals be interpreted, do they majorly reflect post- or presynaptic activity? Do the observed wave trajectories strictly match/follow anatomical cortical connection patterns? Could wave events be shadowed by different amounts of contributions from pre- and postsynaptic events?

The signals reflect the volume average of membrane potential of cells expressing the indicator. As presynaptic membranes contribute very little to the plasma membrane area of cortical cells, we expect a negligible contribution of presynaptic membranes potentials to the measured optical signal. We do not comment on conscious processing, but we agree that our title implies that there is consciousness in mice. The rating as conscious state has now been further supported by the added analysis of motion energy in Fig. R3 and Fig. S3. Thus, we think this is a generally accepted notion, and expansion on the philosophical issue (how to define consciousness) would be beyond our intention and the purpose of our manuscript.

Do the authors believe that the observed waves are strictly generated within the neocortex? Unless I overlooked it, the model seems to incorporate solely intra-cortical interactions? Thus, do the authors exclude contributions from subcortical regions?

We do not exclude contributions from subcortical inputs and in this respect the model is biologically unrealistic. However, one might speculate that intra-cortical mechanisms are sufficient for the generation of slow propagating waves.

In how far is the model scalable to larger (e.g., human) brain sizes?

We do not know the answer to this question but would not see a principle limitation of scalability. Perhaps it is more challenging to incorporate biologically more detailed anatomical properties of local regions and inter-area projections into such models.

Line 22, page 2: It should be made clear which references are related to traveling waves found in spontaneous activity patterns and which references refer to stimulus-triggered waves (e.g., Benucci et al. 2007 and Muller et al., 2018). Moreover, Jancke et al., 2004 should be added here, as this study was the first demonstration of subthreshold propagation vs. suprathreshold propagation of population activity using voltage indicator imaging in direct combination with extracellular recordings. As a side remark, please note that the claim of Benucci et al. 2007 to have distinguished traveling waves from a standing wave was called into question by Sit et al., 2009.

We have now clarified the references for spontaneous and evoked waves and cited the Jancke et al., 2004 paper. We also pointed out that wave could reflect cortical sites responding to stimulus at different latencies, as was shown previously in voltage-sensitive dye imaging in V1 (Sit et al., 2009).”

Given the extensive statements about the speed of waves, speed values should be presented as numbers including statistics. Unless I overlooked it, values are displayed only (in Fig. 2).

We have added to the speed values along with descriptive statistics (initially presented only in Fig. 2), see P. 5.

Several figures represent a single experiment/animal without reporting values or performing statistical tests. In this way, the authors' statements heavily rely on the visual inspection of the figures by the reader. This is a major drawback in my opinion and hinders many of the conclusions drawn from the data. In addition, it should become clear in the main text (rather than in figure legends only) whenever single representative examples are described. E.g., line 1-11, page 4: "In both the anesthetized and wakeful states, we observed a maximum frequency power of the cortex-wide population voltage activity ..."

Here, a list of some further examples of statements with missing statistics:

Lines 2-9, page 5: "... we first quantified the propagation direction and speed of the detected waves. The voltage waves propagated in preferred directions at both brain states, but the wave directions are more restricted to the frontal to parietal direction in the anaesthetized state

"The wave propagation speeds showed a broader distribution in the wakeful state (Fig. 2B)."

"The average speed of PVF decreased (Fig. 2C) even though the average speed of large waves increased (Fig. 2C inset) from anesthetized to wakeful states for all mice analyzed (N = 5 mice)."

We have added statistical results to figures (Fig. 1a, Fig. 2c-e, Fig. 3b, Fig. 7a-b) and the corresponding descriptions in the main text where applicable.

Lines 6-28, page 6: none of the claims are supported by statistics.

Line 27, page 6. "This revealed that the top 5 modes are similar between the anaesthetised and wakeful states (Fig. 3C)." – According to this figure, we see instead how much the SVD modes are similar to the reference mode (1st mode of the anaesthetized state). A straightforward correlation analysis between these modes in both states might justify the above-mentioned statement.

Line 28, page 6: "Results are similar for the other four mice" – Where and how is this statement supported?

Following the suggestion from Reviewer 2, we have now quantified SVD with the data from all animals/samples and compared the variance of the modes in different states with statistics (see Fig. 3). The texts have been revised accordingly.

Finally, the connection of the model to the empirical data appears somewhat weak. E.g.:

Lines 13-14, page 13 "As the strengths of the long-range connections increase, the coherence under the awake state increased whereas the coherence under the anaesthetised state does not show an increase (Fig. 7C)." – However, empirical data in Fig. 7D seem to a considerable extent not to match the simulation shown in Fig. 7C?

Also depictions in Fig. 6G (model) and Fig. 5C (data) "Probability of local complex wave number (source, sink and saddle) ..." do not show a convincing similarity.

The purpose of our model is to qualitatively reveal possible mechanisms governing the dynamic transition from anaesthesia to wakefulness. Thus, our model of cortical waves altered by anaesthesia is minimalistic aiming at some biological plausibility (i.e. increased inhibition, reduced excitability and transmission by anaesthesia, and local and inter-area interactions) but without implying biological realism, aiming to providing some mechanistic understanding of the key spatiotemporal dynamical features in the data, rather than fitting the properties observed in the experiment data (the later purpose is also unavailable since our model is an incomplete description of the true mouse brain network).

Comparison of Fig. 7c and Fig. 7d is difficult because Fig. 7c from the model have finer connection strength scale with 7 different connection strengths, while Fig. 7d from data only have two connection strengths as weak and strong. To make the comparison more appreciable, we now combine the connection strength = 0, 5, 10 as weak, and connection strength = 20, 25, 30 as strong. With this rearrangement of the plot, the main conclusion “As the strengths of the inter-area connections increase, the coherence under the awake state increased (right-tailed Wilcoxon signed rank test, $p < 0.05$, $N = 300$ trials) whereas the coherence under the anesthetized state does not show an increase” becomes clearer but remains unchanged from our initial submission. Comparing our minimalistic model to data, we reach the overall conclusion that “inter-area projections play important roles in corticocortical information transmission in the wakeful than in the anesthetized state”, which is unchanged from our initial submission.

In Fig. 6g, we aimed to show that when reducing the barbiturate anaesthesia effect (reducing parameter p from 0.5 to 0, or changing brain states from 1 to 6), the complex local waves become more frequent and thus frames with more than three patterns become dominant, which is a key feature observed in data. In Fig. 5c, we aimed to contrast the awake state with the anaesthetised state in terms of the proportion of frames with different number of patterns, which also shows that frames with more than three patterns become more dominant whereas those exhibiting <3 wave patterns reduced in the awake state. Thus, they show convincing similarity in terms of the increasing number of frames with more complex wave patterns.

We revised the description of model results to make our statement more clearly, to avoid possible confusion of claiming fitting experimental data by the model which we do not make.

Minor

Line 9, page 2: “More recently, a large body of evidence supported the idea that the increasing complexity of cortical activities is a signature of recovery from general anesthesia to wakefulness or conscious states ...”

Does this statement make sense? To me it reads as if an “increasing complexity of cortical activities” is strictly/exclusively bound to such specific transition. Maybe omitting “the” (before “increasing”) helps? Alternatively: More recently, a large body of evidence supported the idea that a signature of recovery from general anesthesia to wakefulness or conscious states is an increasing complexity of cortical spontaneous activities?

Thank you for this suggestion. We have followed the suggestion to change the sentence to “More recently, a large body of evidence supported the idea that a signature of recovery from general anaesthesia to wakefulness or conscious states is an increasing complexity of cortical spontaneous activities.”

Throughout the text the authors state that “long-range connections” play a major role in the generation of more complex pattern dynamics observed in wakeful states. In the literature (particularly when dealing with larger brain sizes) long-range connections are often associated with horizontal connections within a cortical area. Do the authors refer to “long-range” spanned exclusively between different cortical areas? How were the dimensions of “long-range” specified here?

With “long range” we mean “inter-area” and in the model “between non-neighbouring voxels”. We now unify the term to “inter-area” in the revised version.

Related to Fig. 1: “We observed spatiotemporal patterns in the color-coded voltage maps (Fig. 1C, D), with activity waves that appeared more widespread during the anesthetized state (Fig. 1E).” – During transition to wakefulness, there seems a reduction of wave amplitudes? Is this a true effect? Otherwise, it may be helpful to color-scale plots in Fig. 1D individually.

Yes, this is a “true effect” (i.e. systematic and consistent observation), as also seen by the oscillatory wave form seen above as Fig. R1 in response to reviewer #1.

Propagation distance of the waves appears neither quantified in the empirical data nor in the simulated one.

With phase velocity field, we pay attention to the statistics of wave speed and local complex waves. We did not focus on the analysis of wave fronts and propagation distance which may only be meaningful in the case of some large wave events, which can sweep through the whole imaging field of view.

The authors categorize the animals’ state as either wakeful or anesthetized. According to the literature, recovery from pentobarbiturate is quite variable and can take relatively long (e.g., Field et al., 1993, Haberham et al., 1998). What were the concrete times after the bolus injection during which the recorded trials were considered as reflecting wakefulness?

The reviewer correctly states that recovery times from pentobarbiturate is quite variable and for that reason we did not consider the times after the bolus injection but the heart rate and the video monitoring of movements as indicators of the wake state. For details see above. But more importantly and to address the concern that the “recovery from anaesthesia” condition may differ from fully awake state, we added now data and analysis from mice that have not been exposed to anaesthesia for several days.

Furthermore, could the authors find gradual changes (as opposed to only a binary characterization wakeful or anesthetized) of average speed, heterogeneity etc. in their data - similar as stated in their model (Fig. 6)?

Indeed, we find gradual changes with respect to the analysed wave characteristics. This result is now contained in Fig. R3 (Fig. S3).

In Fig. 6 annotation of speed unity on y-axis is missing.

Now added, thank you.

Line 37, page 16 – line 3, page 17: “... but we shall note that in the real cortex and under physiological conditions, increased excitability during the conscious state are likely to be mediated by endogenous neurostimulators such as acetylcholine and amines.”

The authors might substantiate their proposition with a citation: In a recent review Jancke and colleagues demonstrated changes of ongoing brain dynamics based on activation of a single amine receptor type (5-HT_{2A}) and outlined experimental options for such manipulation (Jancke et al., 2021). In addition, the authors show its different effects on spontaneous activity under anesthesia and awake conditions (Azimi et al., 2021).

We have cited the papers suggested, thank you. As we did not find the paper Azimi et al., 2021, we cite a paper in elife Azimi et al. from 2020.

REVIEWER COMMENTS

Reviewer #1 (Remarks to the Author):

Thank for carefully addressing my comments. I think this is an excellent paper and that it is ready for publication. Congratulations to the authors.

Reviewer #2 (Remarks to the Author):

I thank the authors for their edits, and have the following outstanding comments / suggestions:

1) In the abstract, please also inform about sample sizes and statistics/effects and p-value of the key findings. Last sentence in the abstract sounds too definitive. One would tone down this association by eg using 'likely' etc.

2) Introduction

- P.2 typo: Should read "Traveling waves can have different forms, ..."
- a more detailed justification for why these studies need to be done in eg mice other animal models, could be worthwhile.
- P.3 please define 'principal modes' early on

3) Results (P 3)

- l 109 to 131 please provide numbers of mice that were used for the different analyses. Ditto for l 156-183

Please provide quantitative results to support some of the descriptions eg.

- l 127 appeared more widespread
- l 131 in the anesthetized state were more coherent
- l 248-259 'Hopf instability dominates Turing instability', reduction in average wave speed, increased speed of large waves

4) Discussion

- l. 331 - did the authors truly reveal the mechanism of the neural signatures? didn't they rather reveal a plausible mechanism? I suggest to tone down some of the claims here.

- l. 335 "and have important implications for understanding conscious processes of brain functions." - I would highlight in first discussion paragraph that these findings were derived in mice and that they, while not derived in humans, may have potential implications for understanding conscious processes

Reviewer #3 (Remarks to the Author):

In this revision, the authors have addressed many of the points raised in the first round of review. At the same time, however, some revisions raise major points.

The authors appear to have made an error in the speed calculation: the propagation speeds reported in Rubino et al. 2006 are 20-40 cm^*/s , while the speeds reported in this submission are 20-40 mm^*/s . This is an order of magnitude smaller than the speeds reported in the earlier work. This point underscores the importance of carefully evaluating the propagation speeds presented here. Specifically, my point raised in the earlier round of review concerning speeds remains to be

addressed. It is difficult to understand how the complex spatiotemporal patterns reported here could take 3-4 seconds to propagate over the cortical area. Could these be residual effects of anesthesia, and if so, are they different in the awake-only experiments? Further, could these results on propagation speed be affected by smoothing in the phase analysis (raised but not addressed in the first round of review)? Examples of the raw data demonstrating this point and precluding these confounds would be important here.

Second, lines 61-63 now state that “Some of these waves such as those found in (Benucci et al., 2007) may instead reflect cortical sites responding to stimulus at different latencies as shown in (Sit et al., 2009)”. These are important references to the literature; however, it may be important to state more specifically that this consideration applies to the claim of Benucci et al. 2007, as later work (including a paper from some authors of the present manuscript, i.e. Townsend et al., *Journal of Neuroscience*, 2017) has made this distinction in the analysis of single trials.

As stated above, the revisions in this round raise key technical points on this work. The work remains interesting and novel, but important details remain to ensure the accuracy and completeness of this submission.

Reviewer #4 (Remarks to the Author):

The authors did an overall nice job in addressing my previous comments. Congratulations to this outstanding work!

I have just three minor issues left:

1) Somewhat confusing, page 5, line 169, yellow marked text:

“(Fig. 3b, Wilcoxon rank sum test $p < 0.01$, $N = 5$ trials for anesthetized and post woken, $N=19$ trials for fully awake)”

Legend Fig. 3b claims instead calculation of “sample numbers” and also in their response letter the authors wrote “Following the suggestion from Reviewer 2, we have now quantified SVD with the data from all animals/samples and compared the variance of the modes in different states with statistics (see Fig. 3). The texts have been revised accordingly.”

Thus, “ $N=19$ trials” (Page 5, line 169) should be “ $N=19$ animals/samples”?

2) Jancke et al., 2004 (*Nature* 428, 423-6) is now cited in the main text but seems missing in the reference section.

3) Please note also that Jancke et al., *FEBS J*, published online 2021, has recently been issued: *FEBS J*. 2022 Apr;289(8):2067-2084. doi: 10.1111/febs.15855.

Point to point response to REVIEWER COMMENTS

Original text by reviewers is copied in black font; our responses are marked by **green bold font**.

Reviewer #1 (Remarks to the Author):

Thank for carefully addressing my comments. I think this is an excellent paper and that it is ready for publication. Congratulations to the authors.

We thank the reviewer for the encouraging comments and for taking the time to review our manuscript.

Reviewer #2 (Remarks to the Author):

I thank the authors for the their edits, and have the following outstanding comments / suggestions:

1) In the abstract, please also inform about sample sizes and statistics/effects and p-value of the key findings. Last sentence in the abstract sounds too definitive. One would tone down this association by eg using 'likely' etc.

It is our understanding that it is not the style of Nature Communications to present these details in the abstract although we understand that some clinically oriented journals may follow this style. Based on this understanding, we did not amend the abstract with the sample sizes and statistics/effects and p-value of the key findings. Of course, upon editorial request we are happy to do so.

As suggested by the reviewer, we have also added 'likely' in the last sentence for toning down.

2) Introduction

- P.2 typo: Should read "Traveling waves can have different forms, ..."

This has now been corrected.

- a more detailed justification for why these studies need to be done in eg mice other animal models, could be worthwhile.

The experimental approach relies on genetically modified (transgenic) mice carrying a gene for a genetically encoded voltage indicator. From the technical viewpoint, mice are the established standard animal model for these types of studies. From the ethical viewpoint, we think that flies and worms (other model organisms where voltage imaging using genetically encoded indicators is performed) are not suitable to address the research questions that we posed. Mammalians that are evolutionarily and functionally closer to humans may be considered in the future. This could be worthwhile, as the reviewer correctly states.

- P.3 please define 'principal modes' early on

Thank you for this suggestion. We have amended the manuscript with the following definition: “The principal modes of whole cortex-scale waves, defined as the singular value decomposition (SVD) modes of the phase velocity fields,”

3) Results (P 3)

- l 109 to 131 please provide numbers of mice that were used for the different analyses. Ditto for l 156-183

We have now added this information in the revised manuscript.

Please provide quantitative results to support some of the descriptions eg.

- l 127 appeared more widespread

We changed this sentence to “larger oscillation amplitude” which, indeed, would be more appropriate for Fig. 1e. We also performed statistical testing to support the description of larger oscillation amplitude (right-tailed Wilcoxon signed rank test, $p < 0.05$).

- l 131 in the anesthetized state were more coherent

We quantified this result and now provided descriptive statistics in our revised manuscript with the following text:

“Next, we used homogeneity to measure the coherence in the wave propagation direction. We consistently found a significant decrease (right-tailed Wilcoxon signed rank test, $p < 0.05$, $N = 5$ mice) in homogeneity from anesthetized to post woken and fully awake states, indicating that wave propagation directions are more disordered in the post woken and fully awake states (Fig. 2d).”

- l. 248-259 'Hopf instability dominates Turing instability', reduction in average wave speed, increased speed of large waves

We have amended our manuscript with a call out to Fig. 6B which shows clear reduction or increased trend with Cox-Stuart test $p < 0.05$.

4) Discussion

- l. 331 - did the authors truly reveal the mechanism of the neural signatures? didnt they rather reveal a plausible mechanism? I suggest to tone down some of the claims here.

- l. 335 "and have important implications for understanding conscious processes of brain functions. " - i would highlight in first discussion paragraph that these findings were derived in mice and that they, while not derived in humans, may have potential implications for understanding concious processes

Thank you for these suggestions. We have toned down these claims as the reviewer recommended.

Reviewer #3 (Remarks to the Author):

In this revision, the authors have addressed many of the points raised in the first round of review. At the same time, however, some revisions raise major points.

The authors appear to have made an error in the speed calculation: the propagation speeds reported in Rubino et al. 2006 are 20-40 cm^*/s , while the speeds reported in this submission are 20-40 mm^*/s .

This is an order of magnitude smaller than the speeds reported in the earlier work. This point underscores the importance of carefully evaluating the propagation speeds presented here. Specifically, my point raised in the earlier round of review concerning speeds remains to be addressed.

We agree with the reviewer if this concern was prompted by our reference to Rubino et al. 2006; this reference has not been well considered by us. However, our calculation of the speed of the large traveling waves is unlikely erroneous as it is consistent with several independent reports using mice. The confusion caused by our reference to Rubino et al. lies in the fact that Rubino et al. report on observations of waves with high oscillation frequency from primates while we studied mice and our erroneous statement that the values are similar.

Regarding the propagation velocity of slow large waves in rodents: in Dasilva et al. 2021,

it has been shown that the average wave speed in the cortex of anesthetized mouse is 18.2 ± 1.7 mm/s, which increases from deep to light anesthesia levels, as found in our study. In Greenberg et al. 2017, it has also been reported that the average wave propagation speed is ~ 18 mm/s in the mouse cortex. Further, and in line with the above, Greenberg et al had also argued an evolutionary scalable property in the velocity difference across species as a potential explanation (which is beyond the direct scope of the work presented in our manuscript).

Notwithstanding, we fully appreciate that our reference to Rubino et al may have led to this concern from the reviewer and we hope to have now clarified this in our revised manuscript.

It is difficult to understand how the complex spatiotemporal patterns reported here could take 3-4 seconds to propagate over the cortical area.

We would like to highlight that complex waves are local and as such DO NOT propagate over larger cortical area. We also respectfully note that the mouse brain is smaller than the dimensions apparently used for above calculation.

Could these be residual effects of anesthesia, and if so, are they different in the awake-only experiments?

We reported the values recorded without conceivable residual effects of anesthesia (wake animals that were free of anesthesia for several days), see Figures 1-2.

Further, could these results on propagation speed be affected by smoothing in the phase analysis (raised but not addressed in the first round of review)? Examples of the raw data demonstrating this point and precluding these confounds would be important here.

To address this concern, we performed the following analysis: For the complex wave patterns identified based on the PVF (see Fig. R1 A for an example), we calculated their speeds based on amplitude gradients. There is less smoothing involved in this calculation than the phase vector field-based speed, because the latter is based on the Horn-Shunck optical flow method that introduces extra spatial smoothness and continuous constraints. More specifically, we used amplitude gradients to calculate the speed similar to Rubino et al 2006 (they used phase). Let $A(x,y,t)$ be the amplitude of voltage activity at time t and coordinates x and y of the location. We calculated

$speed(t) = \frac{|\partial A / \partial t|}{\|\nabla A\|}$ for complex waves; Fig. R1 B shows that the speed distributions are in the same order for the anesthesia trial (anesthesia trail 2 from mouse 1, 538 sources). We performed the same analysis on fully awake states (fully awake trial 2 from mouse 1, 704 sources); the result is shown in Fig. R1 C & D. These analyses indicate that the effects of anesthesia and smoothing of phase analysis will not affect the speed in orders.

Fig. R1 Comparison of wave speeds calculated by phase velocity fields and amplitude in raw and filtered data.

We have added some explanation and related references in the manuscript, see lines 146-153.

Second, lines 61-63 now state that “Some of these waves such as those found in (Benucci et al., 2007) may instead reflect cortical sites responding to stimulus at different latencies as shown in (Sit et al., 2009)”. These are important references to the literature; however, it may be important to state more specifically that this consideration applies to the claim of Benucci et al. 2007, as later work (including a paper from some authors of the present manuscript, i.e. Townsend et al., Journal of Neuroscience, 2017) has made this distinction in the analysis of

single trials.

Thanks for pointing out this. In the revised manuscript, we have made the referencing more specific by differentiation between spontaneous waves and external-stimulus triggered waves (line 57-60).

As stated above, the revisions in this round raise key technical points on this work. The work remains interesting and novel, but important details remain to ensure the accuracy and completeness of this submission.

We thank the reviewer for the thoughtful comments on our manuscript. We believe the reviewer's major concern on the technical point stems from our less than well considered referencing of results obtained in monkeys and our direct comparison with these data, and we hope to have clarified this misunderstanding.

Reviewer #4 (Remarks to the Author):

The authors did an overall nice job in addressing my previous comments. Congratulations to this outstanding work!

Thank you.

I have just three minor issues left:

1) Somewhat confusing, page 5, line 169, yellow marked text:

“(Fig. 3b, Wilcoxon rank sum test $p < 0.01$, $N = 5$ trials for anesthetized and post woken, $N=19$ trials for fully awake)”

Legend Fig. 3b claims instead calculation of “sample numbers” and also in their response letter the authors wrote “Following the suggestion from Reviewer 2, we have now quantified SVD with the data from all animals/samples and compared the variance of the modes in different states with statistics (see Fig. 3). The texts have been revised accordingly.”

Thus, “ $N=19$ trials” (Page 5, line 169) should be “ $N=19$ animals/samples”?

$N = 19$ trials (i.e. “recordings” as per voltage imaging parlance) collectively from four mice (mouse 1: ten trials, mouse 6: seven trials, mouse 7: one trial, mouse 8: one trial). We have added explanations in the method section to clarify this: “We used one trial as “anesthetized” and one trial as “post woken” in each recovery mouse imaging experiments. Thus, we have a total of 5 trials for anesthetized and 5 trials for post woken

for the analysis. In addition, four mice were analyzed in the fully awake state (mouse 1: ten trials, mouse 6: seven trials, mouse 7: one trial, mouse 8: one trial)”.

2) Jancke et al., 2004 (Nature 428, 423-6) is now cited in the main text but seems missing in the reference section.

Duly corrected, thank you.

3) Please note also that Jancke et al., FEBS J, published online 2021, has recently been issued: FEBS J. 2022 Apr;289(8):2067-2084. doi: 10.1111/febs.15855.

Duly updated, thank you.

REVIEWER COMMENTS

Reviewer #2 (Remarks to the Author):

The reviewers have done a good job at answering my previous comments, but I still have a few open issues

- 1) To stay a bit more balanced, the last sentence abstract could perhaps rather read:
These mechanisms possibly endow the awake ...
- 2) Please give justification for doing this study in mice in introduction
- 3) I previously suggested to the authors that it is worthwhile to highlight in first discussion paragraph that findings were derived in mice. This seems to have been missed in the revision.
- 4) Furthermore, I think it would be good to have a more general paragraph in the discussion to discuss in how far these results can be extrapolated in humans, also acknowledging potential limitations of using mice to understand human consciousness and psychiatric disorders (see eg P. 12)

Reviewer #3 (Remarks to the Author):

In this revision, the authors have largely addressed the previous concerns. Importantly, please note mouse V1 is several millimeters (~4 mm) A-P. With the speeds for complex waves reported here (e.g. 1 mm/s for the individual subject plotted in red, "post woken", Fig. 2c), this seems to clearly imply ~4 seconds for the waves to propagate across the cortical area. While this point should not prevent publication of the manuscript, the authors should carefully check and consider these speeds.

Point to point response to REVIEWER COMMENTS

Original text by reviewers is copied in black font; our responses are marked by green font.

REVIEWERS' COMMENTS

Reviewer #2 (Remarks to the Author):

The reviewers have done a good job at answering my previous comments, but I still have a few open issues

We again thank the reviewer for taking the time to help improving our manuscript.

1) To stay a bit more balanced, the last sentence abstract could perhaps rather read:

These mechanisms possibly endow the awake ...

Thank you for the suggestion. We have changed as suggested.

2) Please give justification for doing this study in mice in introduction

We have now added the following sentence into the introduction:

“Here, we use mesoscopic high spatial coverage voltage imaging approach to monitor cortex-wide activity, which currently has only been achieved in mice. We then use the advanced PVF wave analysis...”

3) I previously suggested to the authors that it is worthwhile to highlight in first discussion paragraph that findings were derived in mice. This seems to have been missed in the revision.

We thank the reviewer for following up on this point.

We have now highlighted in first discussion paragraph that findings were derived in mice: “In this study, by combining empirical observation and modeling, we have found a novel set of signatures for the transition from anesthetized to wakeful states based on travelling waves in mice - including the increased spatiotemporal complexities of localized wave patterns - and outlined the possible underlying mechanism of these neural signatures.”

4) Furthermore, I think it would be good to have a more general paragraph in the discussion to discuss in how far these results can be extrapolated in humans, also acknowledging potential limitations of using mice to understand human consciousness and psychiatric disorders (see eg P. 12)

We have added a paragraph into Line 392-404 on how far these results can be extrapolated in humans. We also now acknowledged the potential limitations of using mice to understand

human consciousness and psychiatric disorders:

“Limited by the practical resolutions in either space or time, human studies using EEG and fMRI often characterize neural spatiotemporal activities with temporal variability (e.g., entropy measure of complexity) or correlation (e.g., functional connectivity). It has been observed that increased entropy and decreased functional connectivity are associated with the emergence of consciousness from sleeping to wakeful state in humans ^{39–41}. A comparison between healthy wakeful human subjects and patients with disorders of consciousness further suggested that conscious cognition could be associated with long-distance coordination and high modularity in functional connectivity ⁵. The current study suggests that these observations in humans could be understood under the travelling wave framework, as the relative dominance of local events could cause increased entropy while reduced global waves could result in decreased functional connectivity. However, caution should be exercised when generalizing the results from mice to humans ⁴².”

Reviewer #3 (Remarks to the Author):

In this revision, the authors have largely addressed the previous concerns. Importantly, please note mouse V1 is several millimeters (~4 mm) A-P. With the speeds for complex waves reported here (e.g. 1 mm/s for the individual subject plotted in red, "post woken", Fig. 2c), this seems to clearly imply ~4 seconds for the waves to propagate across the cortical area. While this point should not prevent publication of the manuscript, the authors should carefully check and consider these speeds.

We thank the reviewer for the further suggestions. We have thoroughly checked the speeds of local waves; as we explained in our previous response, such small speeds are mainly due to the existence of local complex waves with singularity (zero speed). As we highlighted in Results, local waves do not propagate across large regions. We agree with the reviewer that careful considerations will be needed in the future to explore the implications of such complex neural waves in neural information processing.